# Aeolus L2A Aerosol Optical Properties Product: Standard Correct Algorithm and Mie Correct Algorithm

Thomas Flament[1], Dimitri Trapon[1], Adrien Lacour[1], Alain Dabas[1], Frithjof Ehlers[2], and Dorit Huber[3]

[1]CNRM, Université de Toulouse, Météo-France, CNRS, Toulouse, France
[2]ESA/ESTEC, Keplerlaan, Noordwijk, The Netherlands
[3]DoRIT, Munich, Germany

**Correspondence:** Thomas Flament (thomas.flament@meteo.fr)

**Abstract.** Aeolus carries ALADIN, the first High Spectral Resolution Lidar (HSRL) in space. Although ALADIN is optimized to measure winds, its two measurement channels can also be used to derive optical properties of atmospheric particles, including a direct retrieval of the lidar ratio.

This paper presents the Standard Correct Algorithm and the Mie Correct Algorithm, the two main algorithms of the optical properties product called Level 2A product, as they are implemented in version 3.12 of the processor, corresponding to the data labelled Baseline 12. The theoretical basis is the same as in Flamant et al. (2008). Here, we also show the in-orbit performance of these algorithms. We also explain the adaptation of the calibration method, which is needed to cope with unforeseen variations of the instrument radiometric performance due to the in-orbit strain of the primary mirror under varying thermal conditions. Then we discuss the limitations of the algorithms and future improvements.

We demonstrate that the L2A product provides valuable information about airborne particles, in particular we demonstrate the capacity to retrieve a useful lidar ratio from Aeolus observations. This is illustrated using Saharan dust aerosol observed in June 2020.

## 1 Introduction

The Aeolus satellite from the European Space Agency was launched on 22 August 2018, after a long development phase. Carrying a single instrument, the Doppler lidar ALADIN (Atmospheric LAser Doppler INstrument), Aeolus is the first satellite that can measure vertical profiles of wind at the global scale, from the surface of the Earth up to the lower stratosphere (20 km to 25 km of altitude depending on the settings). The lidar operates in the UV at the wavelength $\lambda = 354.8nm$. This short wavelength was chosen in order to enhance the molecular backscatter and allow measurements at high altitudes where aerosols and clouds are scarce and cannot serve as wind tracers. The lidar was designed and optimized for the measurement of the wind. With its two channels, it is also what is usually called a high-spectral resolution lidar (Shipley et al., 1983). The two separate main optical detection channels on board ALADIN are referred to as the Mie and Rayleigh channels. Both channels actually detect a mixture of particulate and molecular scattering. The Mie channel is more sensitive to the spectrally narrow return from atmospheric hydrometers (Full Width at Half Maximum (FWHM) is on the order of tens of MHz), while the Rayleigh channel primarily detects the spectrally broader backscatter from atmospheric molecules (FWHM of several GHz) (Dabas et al., 2008).

In the Rayleigh channel, the detection efficiency of photons backscattered by molecules is about two-times better than for particle-backscattered photons while the Mie channel does the inverse with a sensitivity 30% better for particle-backscattered photons. With a precise calibration of the instrument, the number of photons backscattered by both types of target can thus be separated. This allows the independent measurement of the backscatter and extinction coefficients of aerosols or clouds (Flamant et al., 2008; Ansmann et al., 2007), and thus provides a direct measurement of the extinction to backscatter ratio

called the "lidar ratio". This ratio gives an additional piece of information on the type of particle (Ackermann, 1998; Noh et al., 2007; Yorks et al., 2011; Illingworth et al., 2015; Shen et al., 2021). CALIPSO (Winker et al., 2007) is a lidar mission that has been in space since 2006 and has produced an immense set of vertically resolved profiles of optical properties of aerosol and clouds. In the CALIPSO algorithms, the type of target is estimated using information from several wavelengths and a depolarization channel, a default lidar ratio is specified for each type and extinction and backscatter coefficients are

then inverted (Omar et al., 2009). In Aeolus, a specific algorithm has been developed to exploit the high spectral resolution capacity of ALADIN. It produces the Level-2A (L2A) product of the mission. It is implemented in the Aeolus ground-segment and delivers data in near-real time. Aerosols impact the climate (Houghton et al., 2001), either directly by absorbing the down-welling visible light from the sun or up-welling infrared radiation emitted by the earth, or indirectly by serving as cloud-condensation nuclei, thus changing the concentration and size distribution of cloud drops, and hence their optical properties and

life cycle (Haywood and Boucher, 2000; Yu et al., 2006; Oikawa et al., 2018). These impacts depend on the size, concentration and nature of the aerosols. In this context, the detection of aerosol plumes from space coupled with an information on their type is most welcome to better understand the interaction between aerosols and the earth climate. Aeolus is a first step towards this goal before the launch of ESA EarthCARE mission (Illingworth et al., 2015) that will also operate a high-spectral resolution lidar in the UV, this time specifically designed for the observation of aerosol and clouds. Studies have also been conducted

on the assimilation of Aeolus L2A data by numerical air-quality prediction models. They suggest they can improve air-quality forecasts(Letertre-Danczak et al., 2021).

    The purpose of this paper is to summarize in a less technical, more easily readable article, what parameters are available in the L2A product and how they are determined (section 2). This paper focuses on the Standard Correct Algorithm (SCA) and Mie Correct algorithm (MCA), which are the historical L2A products developed at Institut Pierre Simon Laplace (IPSL) and

MeteoFrance.

    A first description of the SCA algorithm can be found in Flamant et al. (2008). Since this publication, the software that implements this algorithm (called the L2A processor) evolved while keeping the core principles of the algorithms. Technical aspects of the real ALADIN instrument discovered after launch - the sensitivity of the Rayleigh receiver to the temperatures of the primary mirror of the telescope for instance - had to be taken into account. Additional parameters were also added, like

the lidar ratio or the estimated level of error of the different products and new algorithms are being developed for the future. A detailed, up-to-date presentation of the L2A processor is available with the L2A Algorithm Technical Basis Document (ATBD) (Flamant et al., 2021) that can be can be found on the ESA reference page: https://earth.esa.int/eogateway/catalog/aeolus-l2a-aerosol-cloud-optical-product. This document is updated every time a new version of the L2A processor is integrated in the operational processing chain. In this paper, we introduce the main algorithms of the L2A, discuss the known limitations of

these algorithms and the future improvements that are planned (section 3). A section illustrating L2A products from the main algorithm on a real case follows (section 4). It is followed by a conclusion section.

## 2  The aerosol optical properties product in context of the Aeolus mission

### 2.1  The ALADIN instrument, from the perspective of aerosol observation

A high-level description of the Aeolus mission can be found in Stoffelen et al. (2005) and the ADM-Aeolus science report (ESA, 2008). Between the writing of these documents and the actual launch date, several parameters have changed. Among others, the orbit altitude has been lowered from $400\,km$ down to $320\,km$, the laser emission has gone from bursts of pulses of 7 seconds every 28 seconds (ESA (2008), p. 51) to a continuous mode of operation with a pulse repetition frequency $prf = 50\,Hz$, and the laser energy requirement has been reduced from $120\,mJ$ to $60\,mJ$ (the energy per pulse is around $60 - 70\,mJ$ during the first two years of operation of the second laser). These changes affect the signal to noise ratio and the precision of the measurement: the orbit lowering produces a gain in signal while the lower energy and pulse repetition frequency lead to a loss of signal. The sounding geometry of Aeolus is depicted in Figs. 1 and 2. The line-of-sight of the instrument is directed at $35\,deg$ off-nadir. The signals detected from 20 consecutive pulses are accumulated directly on the detectors. The 20-pulse accumulated signals are transmitted to the ground. They are called measurements and set the finest resolution $\approx 3\,km$, or granularity, of Aeolus products. The sequence of 30 consecutive measurements form a "Basic Repeat Cycle" (BRC). The BRC notion is inherited from the above-mentioned $28\,s$ cycle of the burst-mode operation of the laser. In Aeolus products, the measurements belonging to a BRC are stored in the same data set record and often processed altogether to form a single profile. This BRC profile is also called "observation", and accumulates signal over an horizontal distance of $87\,km$.

Signals detected in the Mie and Rayleigh channels are also integrated vertically in height-bins. The number of height-bins is fixed, equal to 24 on both channels. The Range Bin Settings (RBS), i.e. the altitude and thickness of these bins can be adjusted, separately for each channel. This capacity is used to refine the vertical resolution on particular features like low stratospheric winds in the tropical band for Quasi-Biennial Oscillation studies or extend the observed range higher up to study Polar Stratospheric Clouds, for instance. The RBS can be adjusted up to 8 times during one orbit. The bin thickness is always a multiple of 250 m. Thin bins are used preferably close to the ground where the wind is expected to vary more rapidly. The vertical resolution is relaxed higher up, so that the RBS reaches the stratosphere. Thicker bins are also needed to accumulate enough UV photons when sensing the high atmosphere, where the molecular backscatter coefficient is lower due to the low density of molecules. The RBS is usually set so that there is one or several Mie bins in a Rayleigh bin, so that Mie and Rayleigh signals probe the same volumes. This correspondence is not possible for the top-most Rayleigh bin. Because of a hardware constraint, the top-most Rayleigh bin must start above the top-most Mie bin, the minimum offset is 250 m. Aeolus also has a terrain-following system to avoid loosing too many height-bins below the ground. This provides the best vertical coverage given the technical constraints, but also means the product is not provided on a regular vertical grid, which makes plotting and analysis slightly more complex for the users.

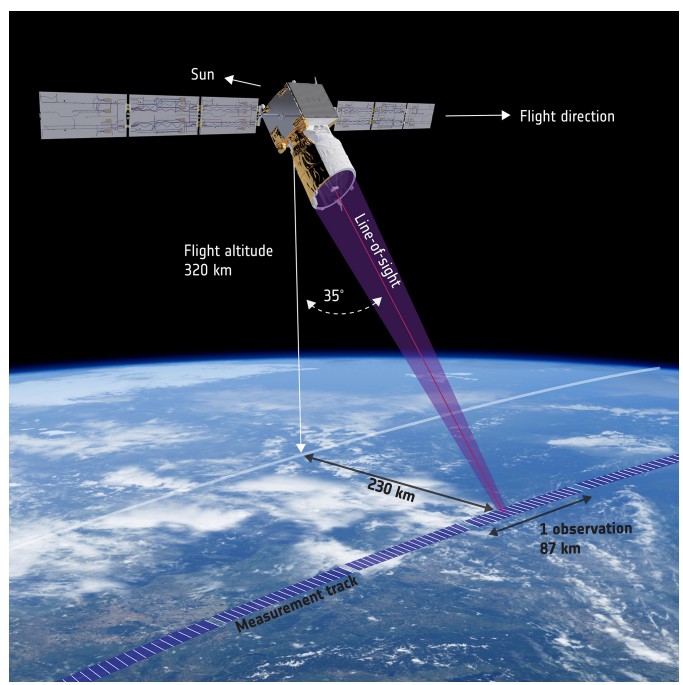

**Figure 1.** Sounding geometry of Aeolus (copyright ESA)

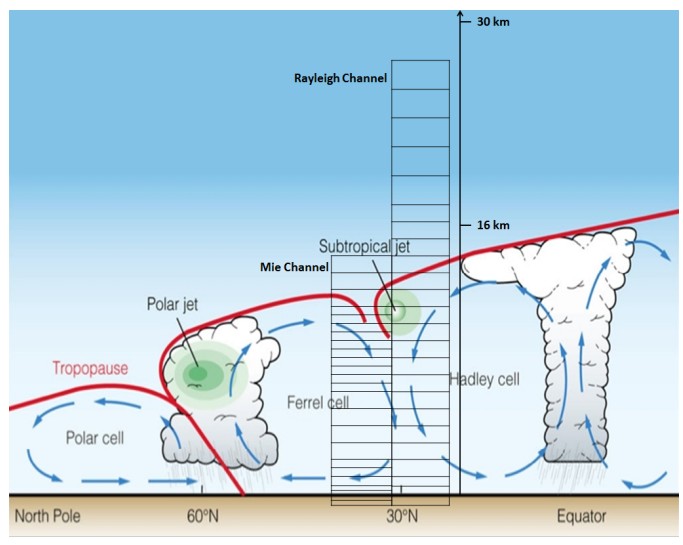

**Figure 2.** Mie and Rayleigh height bins (copyright ESA)

The Rayleigh channel implements a dual Fabry-Perot interferometer. The Mie channel uses a Fizeau interferometer. The transmission curves of the optical filters are drawn in Fig.4, where the frequency axis is given as the frequency offset relative to the emitted frequency. The left panels show the spectral transmission of the two Fabry-Perot interferometers (top) and the Fizeau interferometer (bottom). In both of the left panels, the maximum transmission is normalized to 1, but the overall transmission factor of the Fizeau is $\approx 4$ times less than for the dual Fabry-Perot. The right panels of Fig. 4 shows how an atmosphere-backscattered spectrum (dashed blue line) is filtered by the various interferometers. The Mie peak of the example spectrum is almost completely filtered out by the dual Fabry-Perot as it stands half-way between the peak transmissions of Fabry-Perot A and B, where the sum of the two transmissions reaches a local minimum. In the Mie channel, the Mie peak is almost unfiltered because it stands where the transmission of the Fizeau is at its maximum value. The knowledge of these transmission functions are important for the SCA algorithm. The transmission functions are written as look-up tables in a calibration file, which is described in section 2.2.2.

Overall, photons backscattered by molecules are better transmitted through the Rayleigh channel than the photons backscattered by particles. The intensity of a signal contained in a Mie spectrum after transmission through the Fabry-Perot will be half as much as a Rayleigh-spectrum signal of the same energy at filter input. On the Mie channel, this is the other way around: the energy distributed in a Mie-spectrum is better transmitted than a Rayleigh spectrum.

For the purpose of the L2A, calibration coefficients describing the fraction of light transmitted through the instrument are derived for each channel and for each type of transmitted spectrum (i.e. broad Rayleigh-Brillouin spectrum or narrow Mie peak). $C_1$ is the transmission factor of a Rayleigh-Brillouin spectrum through the Rayleigh channel and $C_4$, the factor for the transmission of a Rayleigh-Brillouin spectrum through the Mie channel. Both factors depend on the energy distribution within the Rayleigh-Brillouin spectrum and so, depend on the atmospheric properties. They are functions of temperature, pressure and Doppler shift. By convention, they are normalized to 1 for "standard" conditions: $1000 hPa$, $300 K$ and no Doppler shift. This normalization means the C coefficients are unit-less. If we try to express this in terms of energy, a backscattered beam with an energy E distributed within a Rayleigh-Brillouin spectrum would yield a signal level proportional to E*C1(P,T,$\delta f$) after going through the Dual Fabry-Perot interferometer and proportional to E*C4(P,T,$\delta f$) after going through the Fizeau interferometer.

$C_2$ is the factor of transmission of a Mie spectrum through the Rayleigh channel and is $\approx 0.5$. $C_3$ is the coefficient of transmission of a Mie spectrum through the Mie channel and is $\approx 1.3$. Both $C_2$ and $C_3$ depend only on the spectral position of the Mie peak, and hence, only on the Doppler shift. They are normalized by the same factors needed for $C_1$ and $C_4$. As the Fizeau transmission windows is narrower, $C_3$ can reach values larger than 1. It translates the fact that some of the energy spread over a Rayleigh-Brillouin spectrum is lost.

## 2.2 Input data of the presented algorithms

The Level 2A makes use of input files that contain, respectively, the L1B data, auxiliary meteorological data and calibration data. Each of these are shortly presented below. In addition, a file containing configuration parameters is needed to run the L2A.

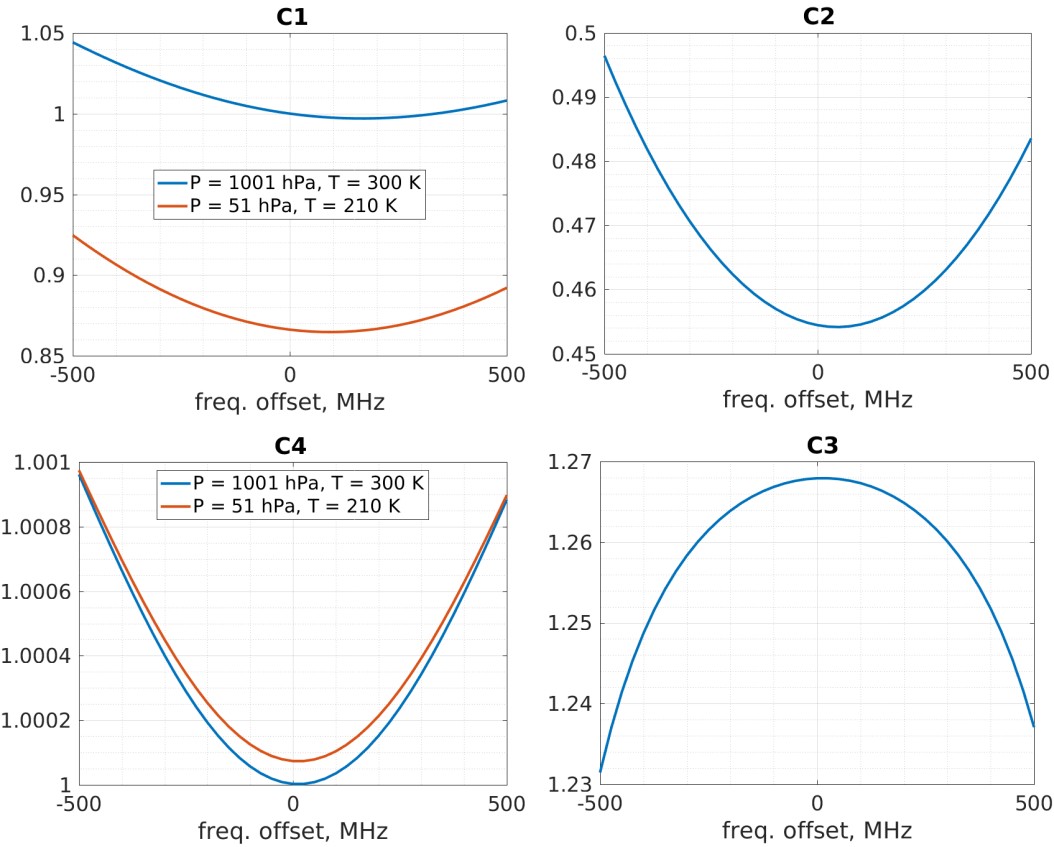

**Figure 3.** Calibration coefficients from an example AUX_CAL. The top panels are transmissions through the Rayleigh channel, bottom panels for the Mie channel. In the left plots, C1 and C4 correspond to the transmission of broad Rayleigh-Brillouin spectra. As these spectra are broad and flat, they do not vary much with Doppler shift. In the right plots, C2 and C3 correspond to the transmission of narrow spectra and are sensitive to local variations of transmissions.

### 2.2.1  L1B data

The basic input of the L2A is the Level-1B (L1B) processor product. A full-description of the processor and the content of the product is given in (Reitebuch et al., 2021). The L1B contains the signals recorded by ALADIN and downlinked to the Earth, corrected from several noise sources or instrumental effects (removal of detector offsets or background light, shadowing
effect of the secondary mirror of the telescope in the image of the Mie fringe on the Mie Charge Coupled Device (CCD) ...). Several variables are derived from these signals. Among them are the Useful Signals, that are the sum of the backscattered signal strength in Mie and Rayleigh bins and the Signal to Noise Ratio of each Useful Signal.

The L1B can also estimate the scattering ratio $\rho = 1 + \beta_p/\beta_m$ where $\beta_p$ and $\beta_m$ are the particulate and the molecular backscatter coefficients. The interference fringe on the Mie CCD is imaged across 16 columns of pixel, corresponding to 16

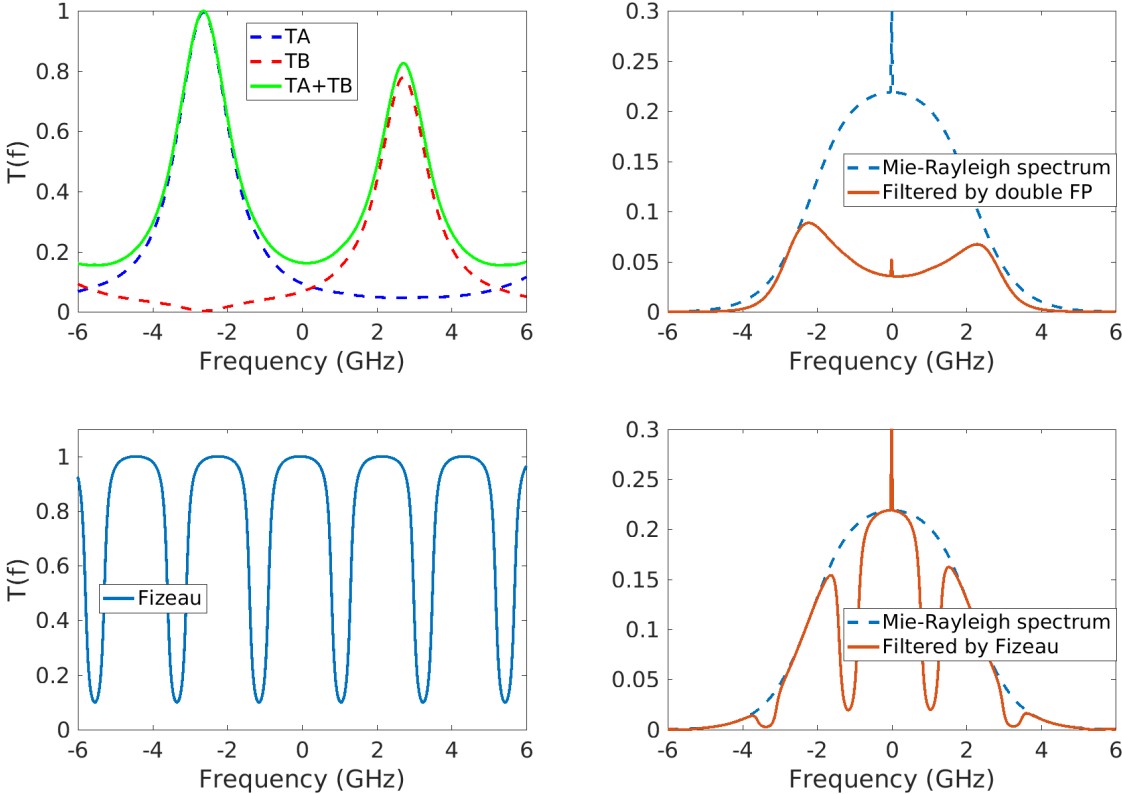

**Figure 4.** Spectral transmission of the two Fabry-Perot (top-left) and Fizeau (bottom-left) interferometers, and filtering of a Mie-Rayleigh spectrum by the double Fabry-Perot (top-right) and Fizeau (bottom-right). The Rayleigh-Brillouin component of the spectrum is for a temperature $T = 300\,K$ and a pressure of $P = 1000\,hPa$). The Mie contribution of the particulate backscatter is the narrow peak on top. The spectrum used here corresponds to a scattering ratio of 1.0025, i.e. it has a very small Mie peak, for illustration purposes.

"sub-channels" of $\approx 100$ MHz bandwidth (see paragraph 4.2.3 in ESA (2008)). The scattering ratio is estimated from the contrast of the image on the CCD of the Mie fringe inside the Fizeau relative to the flat background proportional to the amount of light backscattered by the molecules (the better the contrast, the higher the scattering ratio). In the L2A, this scattering ratio is used to detect bins with a significant amount of particles.

Rayleigh and Mie channel signals for orbit file 10568 are shown in Fig. 5. The cross-talk, i.e. the imperfect separation of molecular and particulate contributions between the two channels, is visible as the Mie channel signal contains a molecular background (dark blue at high altitudes to lighter blue or greenish in the troposphere). Cloud tops are visible in the Rayleigh channel signals as strong signals appear on top of the the columns that they overshadow (yellow features on top of blue columns).

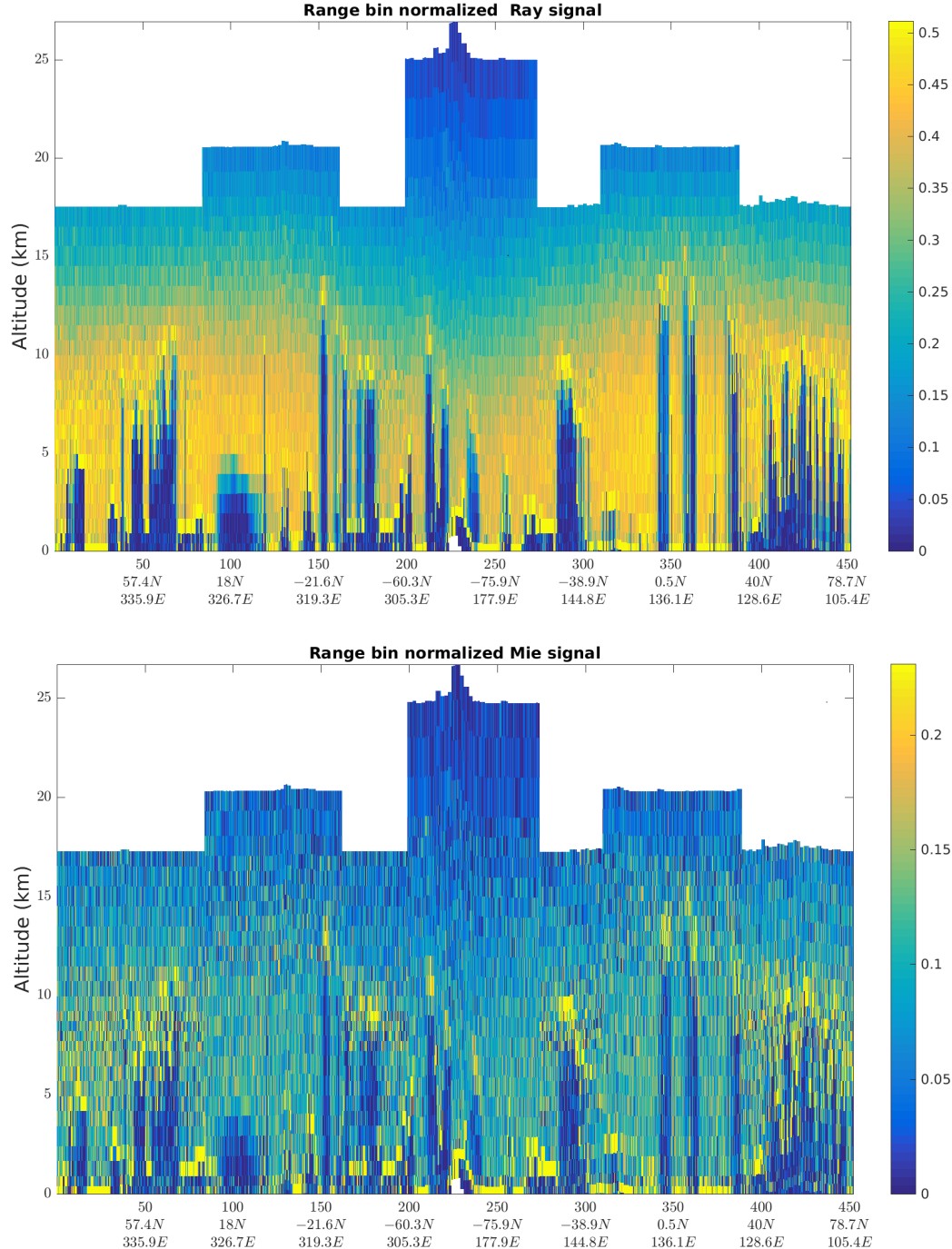

**Figure 5.** Input signals from the L1B product on orbit file 10568. Rayleigh channel (top) and Mie channel (bottom).

### 2.2.2 Calibration data

The calibration data file (AUX_CAL) contains tables of the "C coefficients" introduced above in section 2.1 and appearing in equations 1 and 2 below. They describe how the impact of the pressure $P$, temperature $T$ and Doppler shift on the radiometric efficiency of the receiver. The way they are evaluated is fully described in (Dabas, 2017). They are based on data acquired during two special calibration modes of operation of the lidar. One is called the Instrument Spectral Registration (ISR) mode. During the ISR, the frequency of the laser is swept across a full Free-Spectral-Range of the Fabry-Perot interferometers, i.e.

11 GHz. The signals recorded in the internal path (optical path where a small fraction of the laser pulse is sent directly to the interferometers before going out through the telescope) then describe the monochromatic transmission curves of the Fizeau and Fabry-Perot interferometers. They are used to characterize the interferometers and monitor their evolution. The second calibration mode used for the determination of the $C_i$ coefficients is the Instrument Response Calibration (IRC) mode. In this mode, the whole satellite rotates so that the beam is now vertical, and the frequency of the laser is swept across a smaller

frequency interval of $1 GHz$. Because the beam is vertical, the Doppler shift is assumed to be negligible, and the response of the interferometers to the light backscattered by the atmosphere for each frequency is recorded. Theses responses are then used to try to characterize the differences between the interferometric transmissions in the internal and atmospheric paths (the latter taking into account the telescope and the impact of the field stop inserted in the path in order to limit the amount of background light as much as possible).

In addition, the AUX_CAL contains the radiometric calibration coefficients $K_{Ray}$ and $K_{Mie}$. These are the "scaling factors" that account for the radiometric performance of each channel (see Eq. 1 and 2). As $C_1$ and $C_4$ are normalized to be 1 for standard conditions, $K_{Ray}$ and $K_{Mie}$ describe the capacity of the instrument to transmit light, from the input at the telescope to the output after sensor read-out and conversion to digital values. They are based on the analysis of ISR and IRC data. In section 2.3.2 below, we will see that these valued are superseded by $K_{Ray}$ and $K_{Mie}$ values estimated directly from L1B data

as real data have shown that $K_{Ray}$ and $K_{Mie}$ are different when the lidar is pointing vertically as in the IRC.

### 2.2.3 Auxiliary meteorological data

Because the algorithms make extensive use of the predicted optical properties for a purely molecular atmosphere, we need information about the temperature and pressure of the atmosphere being probed by ALADIN. Pressure and temperature indeed set the density of air, hence the molecular backscatter and extinction coefficients, and change the shape of the backscattered

Rayleigh-Brillouin spectrum.

This information comes in a file of auxiliary meteorological data, called AUX_MET, that is provided by the ECMWF (Rennie et al., 2020). In addition, the file contains horizontal winds, humidity and water content from the model. It contains a forecast for 30 hours of Aeolus track and is provided every 12 hours. The meteorological information spans all of the ECMWF model levels and reach up to 80 km altitude.

## 2.3 L2A product overview

This is a presentation of the product structure and an overview of the algorithms. The core of the SCA and MCA algorithms are consistent with what is described Flamant et al. (2008).

As the processor evolves, its version number is incremented. This paper describes the processor version 3.12. The processor are then picked up by ESA and integrated to the processing facility. The configurations of this facility are labelled as Baselines: each change of processor or major change in processors configurations would trigger an increment in the baseline number. This paper uses the L2A configuration for Baseline 12, which came into Near Real Time production at the beginning of 2021 and was released to the public on 12 May 2021.

Data is downlinked from the satellite at two ground stations, one in Svalbard and one in Troll, Antarctica. This is reflected in the product files that are cut at the downlink time. Most of the files are full orbits, e.g. from Svalbard to Svalbard, but some files only cover half orbits, e.g. from Svalbard to Troll while others files cover more than one orbit. These are often indiscriminately, and improperly, called "orbit files" within the Aeolus DISC consortium.

The L2A product is subdivided in data sets, providing retrievals from several algorithms and ancillary information (quality indicators, geolocation, etc.). This is a quick introduction, before each algorithm and data set is described in more details in its own paragraph.

The first data set contains the geolocation information. It is copied directly from the lower level 1B product (L1B). The L1B contains the lidar signals in a ready-to-use form. For instance they are corrected from the electronics drifts and solar background (Reitebuch et al., 2021). For the sake of traceability, some geolocation information that is specific to the L2A algorithms (e.g. the "mid-bin" altitudes of the SCA) are also written in other data sets.

The main optical properties data sets are derived by the Standard Correct Algorithm (SCA) and the accompanying Product Confidence Data (SCA_PCD). The SCA is described below in section 2.3.1.

The Mie Correct Algorithm (MCA) was designed to use the Mie channel only, originally intended as a back up if the bin matching between both channels is not met. It provides backscatter and extinction coefficients when Rayleigh signals are unavailable or when Rayleigh and Mie height-bins do not match. It performs a cross-talk correction based on the L1B-derived scattering ratio $\rho = 1 + \beta_p/\beta_m$, but it does not fully exploit the high-spectral capability of Aeolus and uses a predefined backscatter-to-extinction ratio for the inversion of Mie signals into extinction and backscatter profiles. This ratio is currently fixed everywhere at 0.07 (i.e. a lidar ratio of $\approx 14.3$).

Lastly, two data sets aim at higher vertical or horizontal resolutions, and are mentioned here for the sake of completeness:

- the Iterative Correct Algorithm (ICA) tries to subdivide the height-bins vertically by testing assumptions on their partial filling by particles.

- the Group product accumulates signal over features with large enough Signal to Noise Ratio (SNR) within one BRC before applying the SCA. The threshold is set at a SNR of 3.5 in Baseline 12. The idea is to remove portions of clear sky between features, in a simple attempt to retrieve more meaningful optical properties for the features.

These products are not further described in this article and we do not recommend to use them in their current version.

### 2.3.1 Standard Correct Algorithm

The main product is the Standard Correct Algorithm (SCA). It uses both Mie and Rayleigh channels to derive cross-talk corrected signals, separating particulate and molecular contributions from the signal.

The ATBD (Flamant et al., 2021) provides a full development of the equations below. For the sake of brevity, only an outline of the SCA algorithm is presented here. Only the main features of the algorithm, necessary to understand the subsequent sections, are covered.

The Rayleigh and Mie signals measured by Aeolus can be written

$$S_{ray,i} = K_{ray} N_p E_0 * (C_{1,i} X_i + C_{2,i} Y_i) \tag{1}$$

$$S_{mie,i} = K_{mie} N_p E_0 * (C_{4,i} X_i + C_{3,i} Y_i) \tag{2}$$

where $i$ is the height-bin index (counted from top to bottom, $i = 1$ is the top-most bin), $N_p$ is the number of accumulated pulses
($N_p = 600$ for an observation), $E_0$ is the pulse energy (monitored on-board and reported in the Level-1B product), $C_1$ to $C_4$, $K_{Ray}$ and $K_{Mie}$ are calibration constants, and

$$X_i = \int_{z_i}^{z_{i+1}} \frac{1}{R^2(z)} \beta_m(z) T_m^2(z) T_p^2(z) dR(z) \tag{3}$$

$$Y_i = \int_{z_i}^{z_{i+1}} \frac{1}{R^2(z)} \beta_p(z) T_m^2(z) T_p^2(z) dR(z) \tag{4}$$

are the molecular and particulate contributions to the signals (called the pure molecular and particulate signals), with $z_i$ the
225 altitude of the top of height-bin $i$ and bottom of height-bin $i-1$), $R(z)$ the satellite-to-target range and

$$T_m(z) = exp\left(-\int_{z}^{z_{sat}} \alpha_m(x) dR(x)\right) \tag{5}$$

and

$$T_p(z) = exp\left(-\int_{z}^{z_{sat}} \alpha_p(x) dR(x)\right) \tag{6}$$

are the one-way transmission through the atmosphere, with $\alpha_m$ and $\alpha_p$ the molecular and particulate extinction coefficients.
The cross-talk corrected signals are normalized by range bin thickness and corrected by the squared range to yield the attenuated backscatters shown in Fig. 6. A quick look at these figures shows that the cloud tops were removed from the molecular signal and the molecular background was removed from the particulate signal (bottom).

The calibration coefficients $C_{1,i}$ and $C_{4,i}$ are explained in section 2.2.2 above.

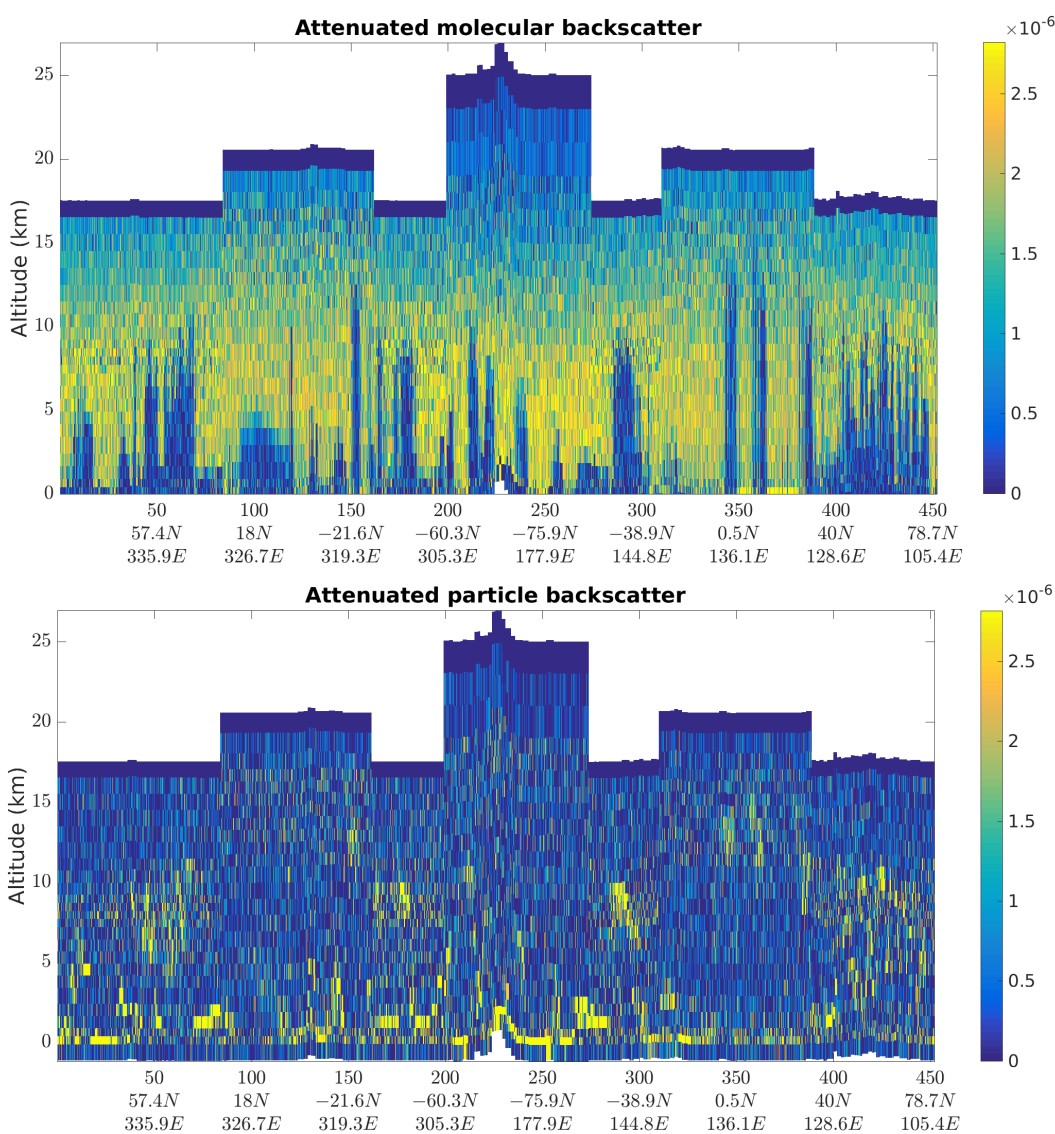

**Figure 6.** Molecular attenuated backscatter (top) and particulate attenuated backscatter (bottom) are shown here on orbit file 10568 at measurement scale, with units $sr^{-1}.m^{-1}$.

The L2A retrieves the pressure $P$, temperature $T$ and line-of-sight wind velocity $v_{los}$ (from which the Doppler shift is computed) conditions inside the height-bins from the AUX_MET file (see 2.2.3).

The determination of the radiometric calibration coefficients $K_{Ray}$ and $K_{Mie}$ is discussed below in section 2.3.2.

Currently, the error on the calibration coefficients is not accounted for in the error estimates that are derived below. The models used in the calibration are not able to describe imperfections of the instrument that were discovered in flight. In particular, the $C_3$ coefficient is derived from simulated transmission curves, and comparison of transmissions and simulations on the internal path show discrepancies of up to 20 %. The actual transmission of the Fizeau interferometer for a light beam backscattered from the atmospheric (the "atmospheric path") is difficult to calibrate. This improved calibration and the inclusion of calibration errors into the error estimates could be addressed in future versions.

In the first operation of the SCA, for each bin $i$, equations (1) and (2) are inverted to:

$$X_i = \frac{K_{mie}C_{3,i}S_{ray,i} - K_{ray}C_{2,i}S_{mie}}{N_p E_0 K_{ray} K_{mie}(C_{1,i}C_{3,i} - C_{2,i}C_{4,i})} \tag{7}$$

$$Y_i = -\frac{K_{mie}C_{4,i}S_{ray,i} - K_{ray}C_{1,i}S_{mie}}{N_p E_0 K_{ray} K_{mie}(C_{1,i}C_{3,i} - C_{2,i}C_{4,i})} \tag{8}$$

This operation is called the cross-talk correction.

From these cross-talk corrected signals, the derivation of the particulate backscatter is straightforward:

$$\beta_p = \frac{Y}{X}\beta_{m,sim} \tag{9}$$

where

$$\beta_{m,sim} = 1.38\left(\frac{550}{\lambda[nm]}\right)^{4.09}\frac{P[hPa]}{1013}\frac{288}{T[K]}10^{-6}m^{-1}sr^{-1} \tag{10}$$

is the molecular backscatter coefficient expected from the pressure $P$ and temperature $T$ information from the AUX_MET file (Collis and Russell (1976)).

For the extinction, the derivation is done recursively from the top of the profile to the bottom. The assumption is made that there are no particles within the first bin and hence that $\alpha_{p,1} = 0$. The L2A uses a so called "Normalized Integrated Two-Way Transmission" (NITWT) defined as

$$NITWT_i = \frac{X_i}{X_1}\frac{X_{1,sim}}{X_{i,sim}} \tag{11}$$

where

$$X_{i,sim} = \int_{z_{i+1}}^{z_i} R^{-2}(z)\beta_{m,sim}(z)T^2_{m,sim}(z)dR(z) \tag{12}$$

with

$$T_{m,sim}(z) = exp\left(-\int_z^{z_{sat}}\alpha_{m,sim}(x)dR(x)\right). \tag{13}$$

$X_{i,sim}$ is the pure molecular signal expected from the $P$ and $T$ profile from the AUX_MET when there are no particles in the atmosphere ($\alpha_{sim} = 8\pi\beta_{m,sim}/3$ as expected from the Rayleigh theory for light scattering by molecules (Collis and Russell, 1976).

The extinction due to particles within bin $i$ is then iteratively determined as:

$$\alpha_{p,i} = \frac{1}{2\Delta R_i} * H^{-1}\left(\frac{1}{T^2_{p,1,i-1}} NITWT_i\right) \tag{14}$$

where $H$ is the function $x \rightarrow H(x) = \frac{1-e^{-x}}{x}$, $\Delta R_i = R(z_{i+1}) - R(z_i)$ is the range thickness of bin $i$, and $T^2_{p,1,i-1}$ is the two way transmission through particles between bin 1 and bin $i-1$. In practice, $H$ is inverted numerically into $H^{-1}$.

The normalization with the signal of the first bin is needed to compensate any extinction that could occur above the topmost bin. But this also makes the SCA extremely sensitive to noise in the first bin. This is discussed in 3.2.

Equations have been derived to estimate the impact of the detection noise on measured signals $S_{Ray}$ and $S_{Mie}$ on retrieved $\beta_p$ and $\alpha_p$ values. The derivation of these error estimates is fully explained in Flamant et al. (2021) but is too cumbersome to be reported here. It is based on the assumption that the uncertainty of $S_{Ray}$ and $S_{Mie}$ is purely due to the Poisson counting noise, and uses second-order developments. As a consequence, error estimates are valid as long as the level of noise is not too high, otherwise the approximation introduced by the second-order developments becomes too coarse. The errors estimates do not take into account the impact of atmospheric heterogeneity within the BRC that increases the random noise on the BRC-accumulation of observation level $S_{Ray}$ and $S_{Mie}$. It nevertheless remains that they are useful to identify the $\beta_p$ and $\alpha_p$ estimations that are reliable, and then give a good idea of their accuracy.

### 2.3.2 Radiometric calibration

Because of thermally induced distortion on the primary mirror (M1), radiometric sensitivity of the instrument varies with the position along the orbit (Weiler et al., 2021). Following the original plan (see Dabas (2017)), calibration measurements are acquired during a special mode, called the Instrument Radiometric Calibration (IRC), during which the satellite is rotated to point at nadir. The objective is to use the molecular atmosphere as a well characterized target, with negligible Doppler shift due to the nadir pointing. This rotation of the satellite causes a change in solar illumination, and results in a different thermal equilibrium of the spacecraft. Although the thermal control loops are performing well, gradients of a few tenths of a degree occur across the primary mirror. Pointing the satellite to nadir also changes these gradients, thus distorting very slightly the M1. This has an impact on the angle of incidence of the beam reaching the interferometers and disturbs the analysis of the backscattered UV light. This change in the instrument characteristics would require a constant recalibration. In the absence of such a correction, both the winds (Weiler et al., 2021) and the aerosol optical properties retrieval would be impacted.

In order to account for these changes, new calibration schemes have been designed. They are based on the evaluation of "clear sky" signals, i.e. signals from a pure molecular atmosphere and above any particulate feature. Such "clear sky" data points are selected by using a threshold (currently 1.16) on the L1B scattering ratio. Any bin below a particle-containing bin a is also rejected. Particles would add an unpredictable amount of backscatter and extinction and spoil signal prediction. In the absence

of particles, the molecular atmosphere provides a predictable target and signals backscattered from these portions of the sky can be accurately compared to predicted molecular returns. We use the auxiliary meteorological data from ECMWF (see 2.2.3) to simulate a purely molecular atmosphere up to 80 km altitude, which encompasses well over $99\%$ of the atmospheric mass. The radiometric calibration coefficients are then the factor needed to scale the predicted signals to the observed ones. Resulting coefficients are shown in Fig. 7 and the impact on the cross-talk correction in Fig. 8.

The first version of the "online" calibration algorithm used the comparison of signals averaged along a full "orbit file". This yielded a single value of radiometric calibration for each channel and each orbit file, which would work for a thermally stable mirror (see the second row of Fig.8). Coefficients are evaluated independently from one orbit file to the next.

Because the thermal distortions of the M1 vary along the orbit, the radiometric performance also vary. The second version of the algorithm (colored line in Fig. 7 and last row of Fig. 8 uses a least square fit of the observed signals to the 12 temperatures $T_i$ read off the thermal sensors of the M1. Radiometric coefficients are then written in the form :

$$K_{Ray} = c_0^{Ray} + c_1^{Ray} * T_1 + C_2^{Ray} * T_2 + ... + C_{12}^{Ray} * T_{12} + \epsilon$$
$$K_{Mie} = c_0^{Mie} + c_1^{Mie} * T_1 + C_2^{Mie} * T_2 + ... + C_{12}^{Mie} * T_{12} + \epsilon \tag{15}$$

This also allows us to interpolate the radiometric calibration in areas where the observed signals are not clear enough e.g. areas where clouds or particles reach too high in the atmosphere for the instrument to collect enough "clear sky" signals.

The successive improvements brought by these calibration techniques are shown in Fig.8. The left panels show the cross-talk corrected signal from particle in "particle-free" bins (other bins are masked in white). It is supposed to be equal to zero. The first correction option (middle row) removes a bias, while the second correction option (bottom row) also removes the large scale patterns that are still visible on the second row (e.g. positive bias around the middle of the orbit). The right panels of this figure show the distribution of the particulate signal in "particle-free" bins corresponding to each calibration option. It shows that the distribution is better centred around 0 after correction, as it is supposed to be. The second version of the radiometric calibration (bottom row) produces a slightly narrower distribution, which means an improvement of the cross-talk correction.

This approach does not account for the residual contamination of supposedly "particle-free" bins and might overestimate the radiometric coefficients by a few percent. Future work will investigate this potential source of bias.

Attenuation by a particle layer that would be above the observable range is also not accounted for. The fit of the signal to the M1 temperatures is made over a full "orbit file", which is at least half an orbit long and most of the time longer, this is enough to guarantee that a high reaching particulate feature in a given region would not bias the fit too much. Algorithms that are able to estimate the total optical depth between the satellite and the top of the highest bin would allow a better characterisation of this flaw in the future (see e.g. Ehlers et al. (2021a)).

### 2.3.3 Mie Channel Algorithm

The Mie Channel Algorithm (MCA) uses observations from the Mie channel alone. It might be useful in places where the extinction retrieval of the SCA does not perform well.

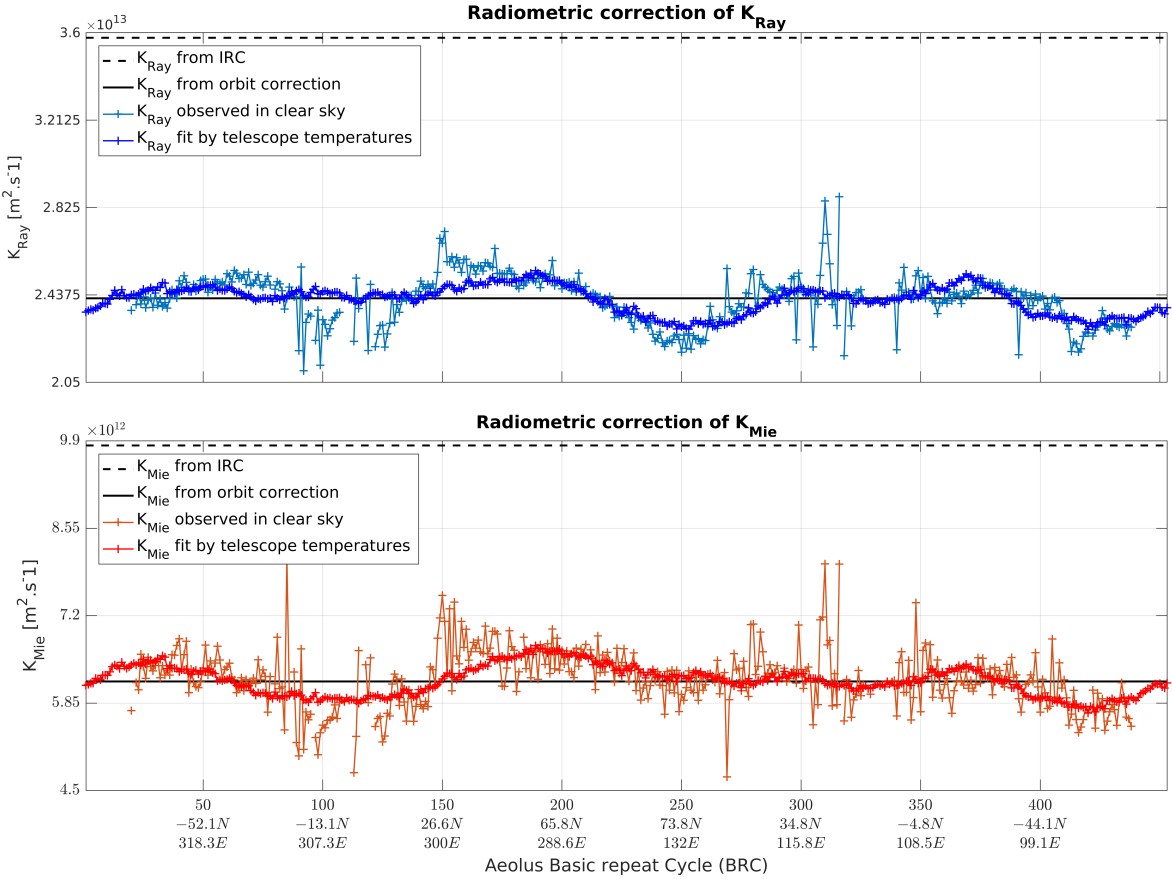

**Figure 7.** Comparison of radiometric calibration coefficients from several algorithms: from the originally planned algorithm using off-nadir acquisition (dashed line), from the orbit averaged online correction (black line) and from the fit with M1 temperatures (bright colored line). They are compared to the local value directly retrieved online from measured clear sky returns (dull colored line, noisier).

The MCA performs a cross-talk correction by using the L1B derived scattering ratio $\rho_{L1B}$:

$$Y_{mie,i} = \frac{S_{mie,i}}{K_{mie} N_p E_0 \left( \frac{C_{4,i}}{\rho_{L1B,i}} + C_{3,i} \right)} \tag{16}$$

It then calls a recursive relation to derive the extinction in a given bin based on the extinction in bins above. It is a version of the Klett algorithm (Klett, 1981).

The initialisation is made in the topmost Mie bin, assuming there is no particulate attenuation between the satellite and
the first bin ($L_{p,sat,1} = \int_0^{R_1} \alpha_p(r)dr = 0$). The molecular attenuation, $L_{m,sat,1} = \int_0^{R_1} \alpha_m(r)dr$ is know from the simulated

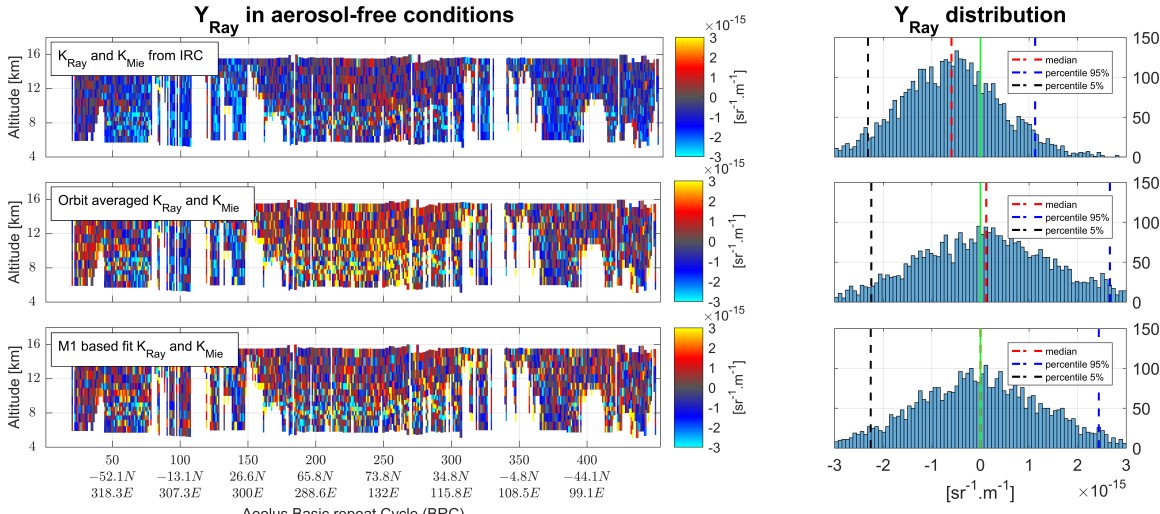

**Figure 8.** Figure showing the evolution of calibration schemes and the resulting improvement of error in signal prediction on orbit #14213 (acquired on 04/02/2021). The left panels show the cross-talk corrected "pure particulate signals" in "particle-free" bins, the histograms on the right show the distributions of these signals.

atmosphere. Then:

$$L_{p,i} = -\frac{1}{2}\ln\left(1 - \frac{2Y_{mie,i}R^2_{mean,i}e^{L_{m,i}}}{T^2_{m,sat,i-1}T^2_{p,sat,i-1}k_p}\right) \tag{17}$$

where $k_p = \beta_p/\alpha_p$ is the inverse of the prescribed lidar ratio and $T_{m,sat,i-1}$ is the molecular transmission to the top of bin $i$ (eq. 6.126 of the ATBD Flamant et al. (2021)).

The MCA major drawback is the use of a prescribed lidar ratio. By fixing this ratio however, the MCA imposes that extinction and backscatter are co-located. This yields a more consistent picture of extinction that is useful to spot features and local variations of extinction. Fig. 9 shows an example of extinction retrieved by the MCA on the orbit file discussed below in section 4.2. It also hints that the dust plume extinction (around BRC 90-110) is under-estimated due to the fixed lidar ratio of 14 (1/0.07) used by the MCA (see Fig. 15 for comparison with the SCA lidar ratio).

As explained in 2.1, the Range Bin Setting (RBS) is changing along the orbit in order to find a compromise between the highest sampled altitude and the resolution. This is visible in the big steps in the maximum altitude of the profile. Smaller steps are due to the terrain following capability , that shifts the RBS to just reach the ground and extend the profile higher up. Some parts of the profiles, especially in the lower atmosphere are not processed (white pixels). This happens when the measured Mie signal becomes negative, often below thick clouds.

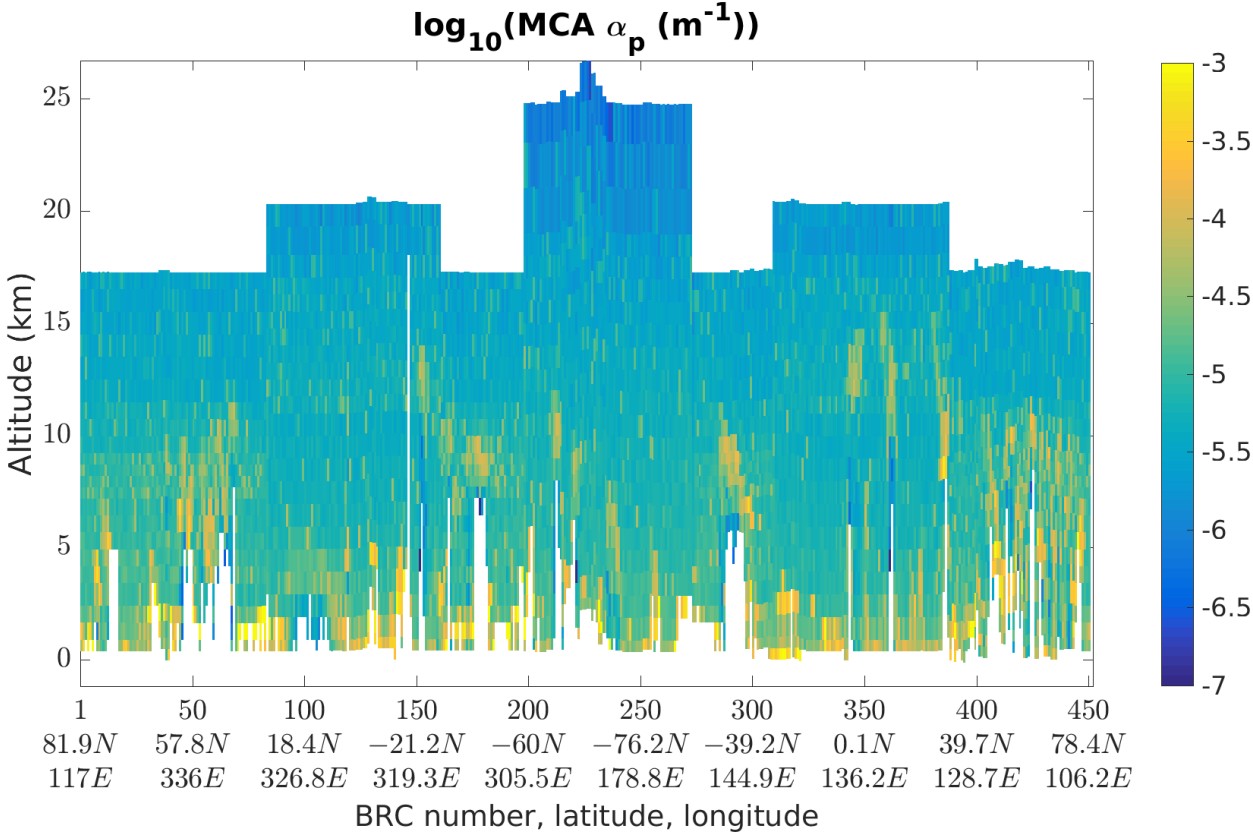

**Figure 9.** Logarithm of the extinction retrieved by the MCA algorithm, on orbit file number 10568, 19 June 2020, starting at 07:51:59 UTC. The dust plume is located around BRC #100 at altitudes below 5 km.

## 3 Known limitations

### 3.1 Instrument limitations

ALADIN was designed primarily for wind determination. The fraction of light sent through the Fizeau interferometer of the Mie channel is smaller than for the Rayleigh channel. The Mie SNR is then lower than the Rayleigh SNR and limits the precision of signals calculated through the cross-talk correction.

Designed as a wind lidar, ALADIN was not initially aimed at observing aerosol optical properties in detail. Under these requirements, it was not fitted with the ability to measure depolarization. The UV laser beam is linearly polarized at the laser output. It goes through a quarter-wave plate (see Fig. 4.13 in (ESA, 2008)) before being routed towards the telescope and is thus transmitted towards the atmosphere with a circular polarization. On the way back, backscattered light goes again through the quarter-wave plate. The circularly polarized light that was transmitted might come back elliptically polarized in the case it was backscattered by depolarizing targets. After going through the quarter-wave plate, it becomes a mix of linearly polarized

light, either along the same direction as the transmitted light (co-polar) or along the perpendicular direction (cross-polar). The beam then reaches a polarizing beam splitter. The co-polar light is routed towards the interferometers, while the cross-polar light is routed back towards the laser and is lost for the analysis. This means that, in order to compare Aeolus observations of backscatter coefficient and lidar ratio to other instruments, only the co-polar component must be considered.

Not going through this extra step before comparing backscatter coefficients would make it seem that the total backscatter of highly depolarizing targets such as ice crystals or dust is largely underestimated by Aeolus.

Because the extinction is derived from signals backscattered by molecules, it is not affected by the depolarization. The co-polar lidar ratio measured by Aeolus on depolarizing targets is going to be larger than the total lidar ratio measured by other instruments.

## 3.2 High noise and extinction retrieval

Extinction can be calculated in a simple way from the molecular backscatter, or more precisely, from its derivative. The SCA is very similar to the classical log-derivative algorithms but the thickness of ALADIN range bins (up to 2 km) mean that the particulate optical thickness ($\alpha_p * \Delta R$) can be large and the approximation used for the molecular extinction (Eq. 6.34 in Flamant et al. (2021)) cannot be used for $\alpha_p$. This is why we later need to inverse function H rather than simply derive the logarithm of the attenuation of the Rayleigh signal. As a side note, this refinement is also the reason why the adjective "correct" was added to the name of the algorithm.

Because we measure extinction by differentiating two bins, the retrieval is very sensitive to noise. We will see that the SCA works well in conditions with high-enough SNR, but it faces its limits in the "real world", where signals have shown to be weaker than expected before launch (Reitebuch et al., 2020).

In order to retrieve the extinction, we compare the observed molecular signal to the molecular signal simulated from the atmospheric conditions (pressure and temperature, from auxiliary meteorological data, extracted from ECMWF NWP forecasts). The ATBD describes in detail how we can access the extinction within bin $i$ from the available observations. The following section intends to give a physical understanding of the SCA extinction retrieval and its limitations.

There are two important aspects to this algorithm: signal in one bin depends on the extinction within all overlying bins and 380 bin $i$ itself. Particle extinction is derived iteratively by propagating downwards the total extinction and comparison to simulated signal. The estimation of extinction is initialized at the topmost bin. By normalizing all molecular signals to the signal in the first bin, we can solve the problem of extinction above the sensing range, i.e. between the satellite and the topmost bin.

But this solution comes with a price: the molecular density, and hence the molecular backscatter, is also the lowest at the top of the sensed atmosphere, usually 20 to 25 km. The contribution of noise is then the strongest in the first bin. The retrieval 385 of extinction by differentiating between the first two bins can yield unrealistically high values. This, in turn, means that the simulated signal is expected to be very low in the next bin. If the extinction was overestimated in the second bin, the signal in the third bin is measured to be larger than expected. This would lead to a negative extinction. If the first extinction is largely overestimated, it can take several bins before the measured signals reach the low expected signal level. This is illustrated in Fig.10, where a large extinction is found in the second bin. This produces a large attenuation on the expected molecular signal

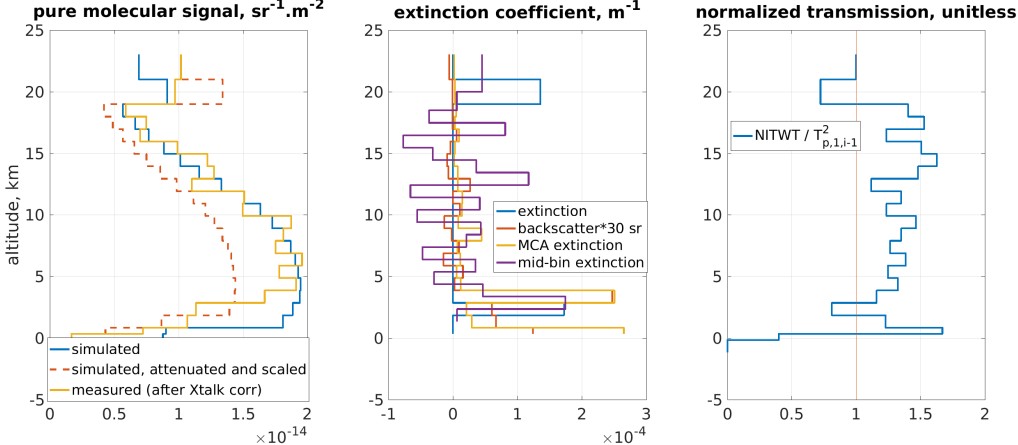

**Figure 10.** Figure illustrating how extinction overestimation in a given bin leads to errors in bins below. The left panel shows observed signals (yellow), simulated signals (blue) and simulated signals with extinction from particles down to the previous bin and normalized by the first bin (dashed red). The extinctions retrieved by several means are shown in the middle panel (SCA in blue, derived from SCA backscatter with an arbitrary lidar ratio of 30 sr in red, from the MCA in red and the mid-bin SCA in purple). The value from which the extinction is derived is shown on the right panel

(red dashed line) which never becomes larger than observed molecular signal (yellow line). The extinction is derived directly from the function on the right panel of Fig. 10, extinction is positive only if this function is below 1. In the given example, we see that because of the overestimation in the first bin, the SCA extinction only yields positive extinction around 2-3 km, whereas the MCA and the SCA backscatter coefficients suggest presence of Aerosols already above. As an indication of the presence of particles, we also show the SCA backscatter scaled by an arbitrary lidar ratio (middle panel, red line). It shows that

the extinction of the SCA (blue line) is detected one bin below the actual particle feature. The MCA extinction is quantitatively wrong because of the fixed lidar ratio, but is detected in the correct bin.

In the SCA, all bins that would have had negative extinction are reset to 0. It is justified by the error propagation in the iterative process. The large error in the retrieval of the first extinction would propagate downwards and create the stripes visible in Fig.11. Resetting negative extinction to 0 allows to reset also this propagation of error, but this comes at the price of bias,

i.e., that the SCA extinction variable "reacts" delayed along the vertical and is unable to detect extinction in bins with an optical depth lower than the cumulated optical depth above.

The propagation of errors through the algorithm, from the signals to the optical properties, also shows the limitations of the SCA extinction retrieval. This is detailed in the ATBD (see eq. 6.82 in Flamant et al. (2021)) and can be summed up by the following formula:

$$\sigma^2_{L_{p,i}} = \approx 4\sum_{k=1}^{i}\langle e^2_{X_k}\rangle - 3\langle e^2_{X_i}\rangle - 3\langle e^2_{X_1}\rangle \qquad (18)$$

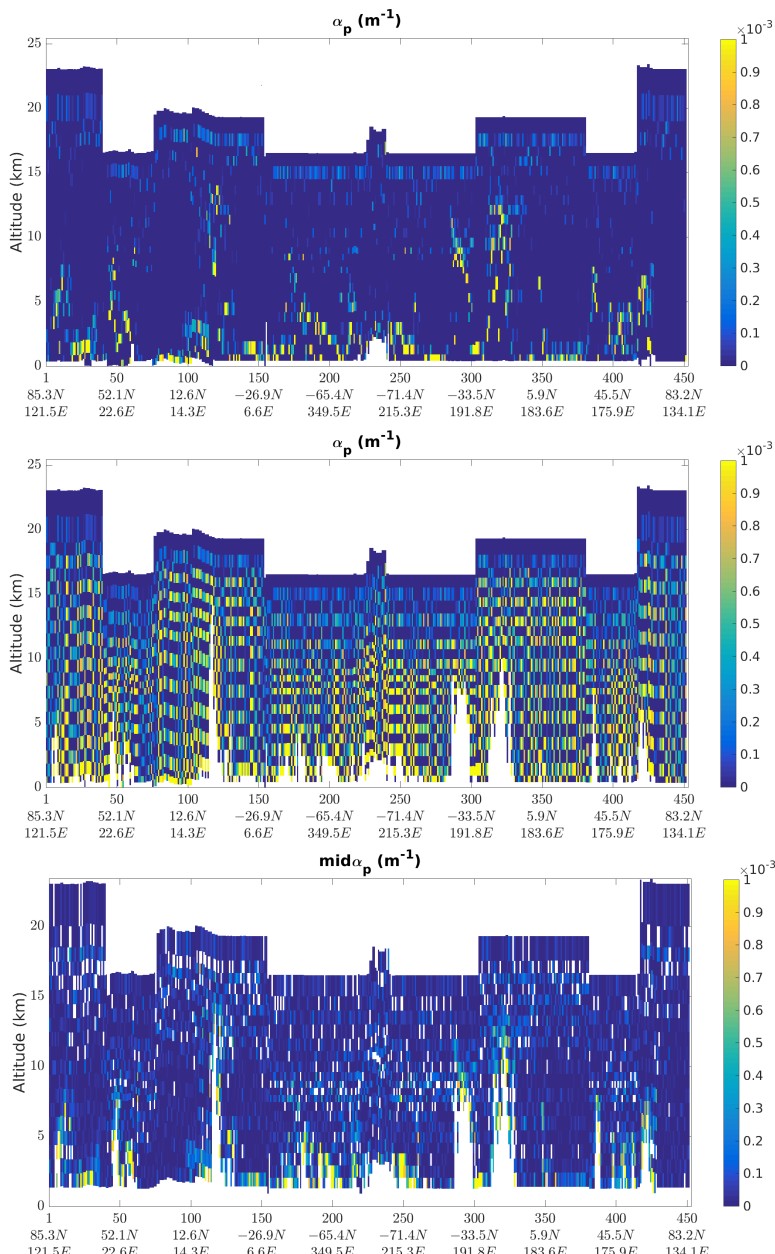

**Figure 11.** Figure showing the extinction retrieved with the SCA (top) and with a modification that cancels the reset of negative values to 0 in the iterative retrieval process (middle). The resulting oscillation due to the downward propagation of errors is obscuring any useful information. The mid-bin retrieval is shown in the bottom panel.

where $\sigma_{L_{p,i}}$ is the standard deviation on particles optical depth in bin $i$, and $e_{X_i}$ is the error added by the observation $X_i$ on top of the actual value $\overline{X}_i$, modeled as $X_i = \overline{X}_i * (1 + e_{X_i})$.

Following Eq. 18, it is not recommended to use the direct computation of extinction on the normal vertical scale, as the error in the first bin is propagated to all of the underlying bins. This drawback can be compensated by averaging two bins. This is done in the "mid-bin" part of the SCA data set. For users interested in extinction properties, it is highly recommended to use the "mid-bin" retrieval of the SCA.

The middle-bin algorithm allows negative values and averages the extinction values over two consecutive bins. The loss of vertical resolution is compensated by a substantial improvement in errors (eq.8.86 of Flamant et al. (2021)). The variance of the retrieved local optical depth becomes:

$$\sigma^2_{L_{p,i}+L_{p,i+1}} = \frac{1}{4}\langle e^2_{X_{i+1}}\rangle + \frac{9}{4}\langle e^2_{X_i}\rangle \tag{19}$$

This estimated standard deviation $\sigma_{L_{p,i}+L_{p,i+1}}$ is no longer linked to $e_{X_1}$, the error on $X_1$, but only to the error in the two bins that are combined to obtain the "mid-bin" value.

In order to compute the lidar ratio by using this "mid-bin" extinction, the "mid-bin" backscatter is also derived by averaging the backscatter in two successive bins.

## 4 Example cases

### 4.1 Simulated example

The Aeolus End-To-End Simulator (E2S) has been developed before the satellite launch to help the development of the ground segment processors (Reitebuch et al., 2018). It simulates the ALADIN instrument in its nominal conditions and generates the downlinked data used by the processor chain. The simulator also takes into account several noise sources including Poisson counting noise at the detector level. The following example is a scenario which is representative of tropical conditions with high convective clouds. The scene is constituted of 240 profiles which match the structure of the Aeolus observation with 8 BRCs of 30 measurements each. This scenario presented in Fig. 12, is an heterogeneous scene with optically thick features that are a challenge for the L2A processor to retrieve the optical properties.

In order to study the sensitivity of the L2A product to noise, 20 independent E2S simulations are run from the same input scene. The noise generated in each simulation is different and, looking at the average retrieval and the standard deviation around it, we can estimate the impact of noise separately from the potential biases of the algorithms. Figure 13 presents how the backscatter and extinction coefficients derived from the SCA mid-bin algorithm compare with the E2S inputs. Most of the time, the backscatter and extinction coefficients are correctly derived, which can be seen where the red line (20-run average retrieval for a given BRC) is close to the black line (E2S input averaged over the BRC). Error estimates for the extinction (brown range around the red line) lie within the range of atmospheric heterogeneity (thin black line, derived as the standard deviation of the E2S input over a given BRC). Backscatter coefficients are also mostly correct, although with a slight low bias, e.g. in BRC 5 between 14 and 10 km altitude (compare the red line to the input, black line). The average of the 20 simulation overlap the expected values with a low dispersion (brown-shaded area) meaning that one realization should be enough to characterize the atmospheric optical properties.

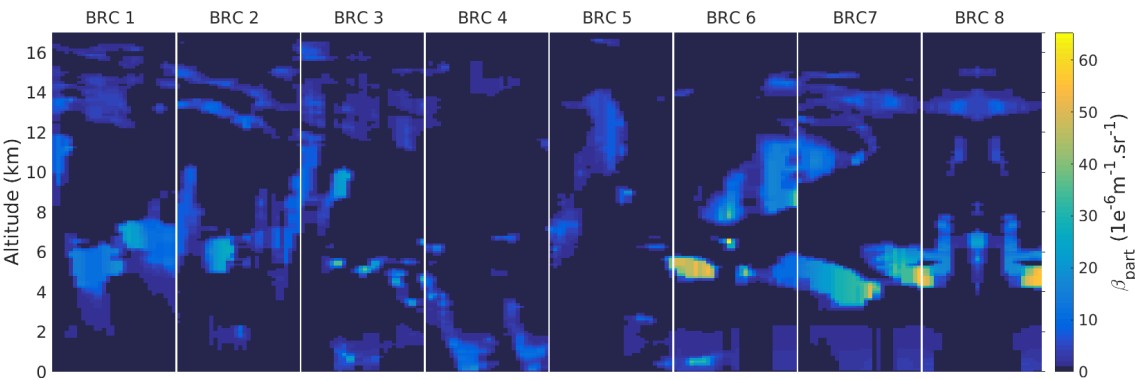

**Figure 12.** Particulate Backscatter input for the E2S. The white vertical lines separate the BRC.

The largest discrepancies are seen below 4 km. In the low levels, the signal can be attenuated by optically thick features above. In addition, the bins below 2 km are only 250 m thick: shorter accumulation time means that signals are lower and that the noise can become dominant. In practice the vertical resolution of the bins is seldom below 500 m. One may notice that some values of the retrieved coefficients are off scale in Fig. 13, they correspond to negatives values. Where noise dominates, the accumulated signal oscillates around zero and the average of the 20 realizations can be slightly negative.

In the BRC 4, the only one in near clear sky conditions, the averaged profile renders the backscatter and the extinction profiles correctly almost all the way down to the last bins. Nevertheless, a small systematic bias is visible when the bin height is 250 m, because the noise is large enough to have large negative values which are cut by the algorithm and not included in the average.

     The BRCs 6 to 8 contain the optically thicker clouds and present the biggest heterogeneities of the scenario. They are also
the BRCs with the less accurate SCA results; the profiles are consistently underestimated between 6 and 10 km, compared to the input profiles averaged over the BRC. The estimated errors are also too low and do not cover the expected values. This is because the variability in these BRCs is mostly due to the heterogeneity within the horizontal accumulation length, which is not taken into account in the error propagation.

     In Fig. 13, the backscatter errors are underestimated by a factor 3 i.e. the light brown area representing the estimated errors
is smaller than the light orange area representing the scatter of retrievals over 20 independent simulations. On the same figure, we see that the extinction errors are reasonably estimated. The bias in the errors can be traced back to the variability of the useful signal in the Mie and Rayleigh channels. In this scenario, the Poisson noise represents only a third of the simulated noise in the Mie channel but is the main contributor for the Rayleigh channel simulated noise. The underestimation of the Mie SNR is then propagated through the derivation to the estimated variances of the backscatter and extinction. To summarize, the
accuracy of the error estimation depends on the significance of the Poisson noise contribution to the signal variability. Factors like the heterogeneity within the observed BRC are not accounted for and lead to an underestimation of the errors by the SCA algorithm.

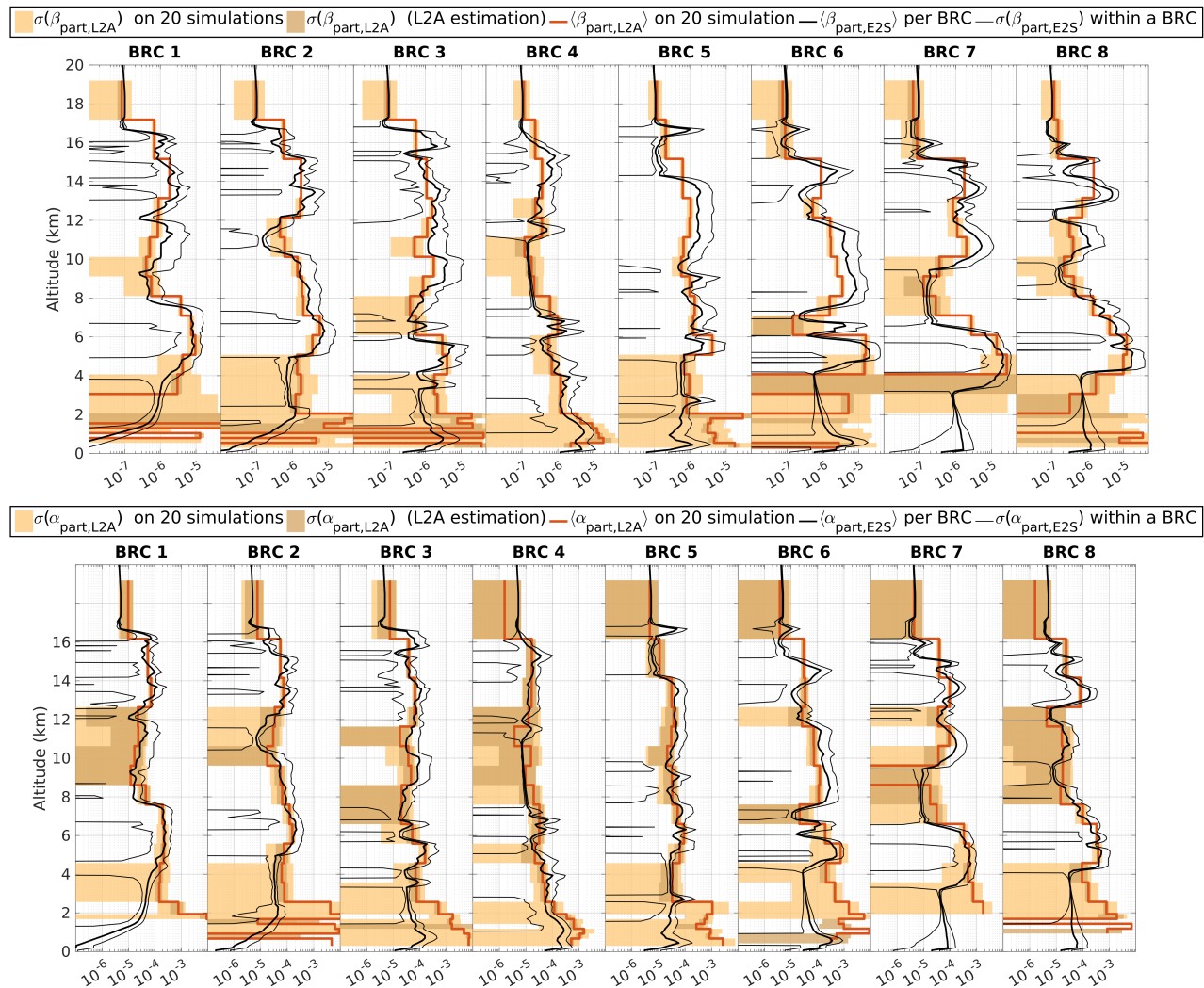

**Figure 13.** Backscatter (Top) and Extinction profiles (Bottom) comparison between E2S simulations processed up to the SCA mid-bin product and the E2S inputs. The black line is the simulation input parameters averaged within each BRC. Thin black line represents the associated standard deviation. Red profiles are the mean coefficients retrieved by the SCA algorithm from 20 realizations. Lighter shading is the associated standard deviation i.e. the true variability. Darker shading delimits the mean error estimated by the SCA error propagation algorithm.

This simulated example proves that, with the correct calibration coefficients, the SCA algorithm is able to render the clouds and aerosols optical properties of a complex scene and provide reasonable error margin as long as the Poisson noise is the main source of noise.

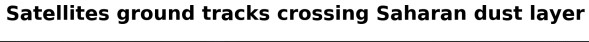

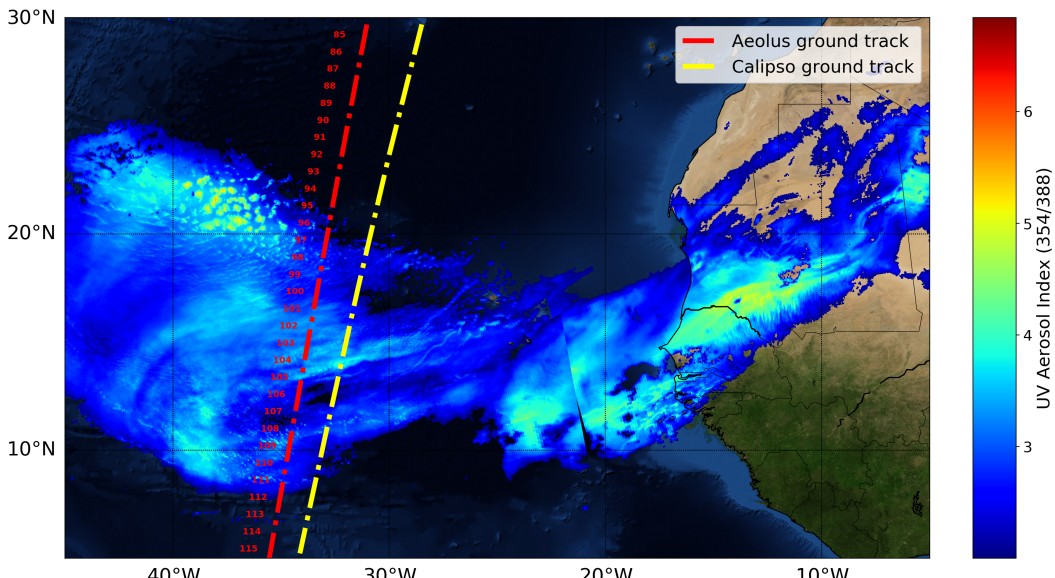

**Figure 14.** Figure showing west African coast and North east Atlantic Ocean from ArcGIS World Imagery from Online basemap (Esri et al., 2021) overlaid with Aeolus (red dashed line) and CALISPO (yellow dash-dotted line) ground tracks and BRC number between 8:08 to 08:16 UTC and Sentinel-5P UV Aerosol index from 388 and 354 nm spectral bands

## 4.2 Saharan dust across the Atlantic ocean in June 2020

On June 2020 numerous instruments such as TROPOMI aboard Copernicus Sentinel-5P (Zweers, 2018), VIIRS aboard NOAA Suomi NPP (Jackson et al., 2013) or CALIOP aboard NASA/CNES CALIPSO (Winker et al., 2009; NASA, 2021) observed a massive dust plume ejected from Saharan Desert to the Atlantic ocean. The brown plume was highly visible from 13 to 20 June with true reflectance images. Aerosol Index (AI) product also showed the presence of UV-absorbing particles (e.g. OMPS aboard NOAA Suomi NPP observed AI up to 3 for the thickest part of the dust plume). Therefore this scene was selected in the frame of the evaluation of L2A product quality using external/validation data.

On 19 June 2020 Aeolus crossed the North Atlantic Ocean between 08:00 and 08:30 UTC and the ALADIN instrument ground track (i.e. Aeolus displayed ground track is the intersection of ALADIN laser slant line of sight pointing 35° offset from nadir with the ground) intersected the western portion of the Saharan Air Layer (SAL) for approximately 20 BRCs. On the same day the dust layer can be seen with Aerosol index derived using the 388 and 354 nm spectral bands produced by two TROPOMI overpasses from 14:50 to 16:32 UTC and from 13:09 to 14:50 UTC (Sentinel-5P (2021), downloaded on 23 Feb. 2021). TROPOMI UV aerosol index product for the core of the plume (i.e. UV AI ≥ 2) is shown in Fig.14 overlaid with Aeolus and CALIPSO ground tracks.

The SCA lidar ratio in Fig. 15 is read from the mid-bin product and quality flags have been applied. According to these flags, the backscatter coefficient retrieval is considered as valid in a specific bin if the Mie SNR is larger than 40 and the extinction

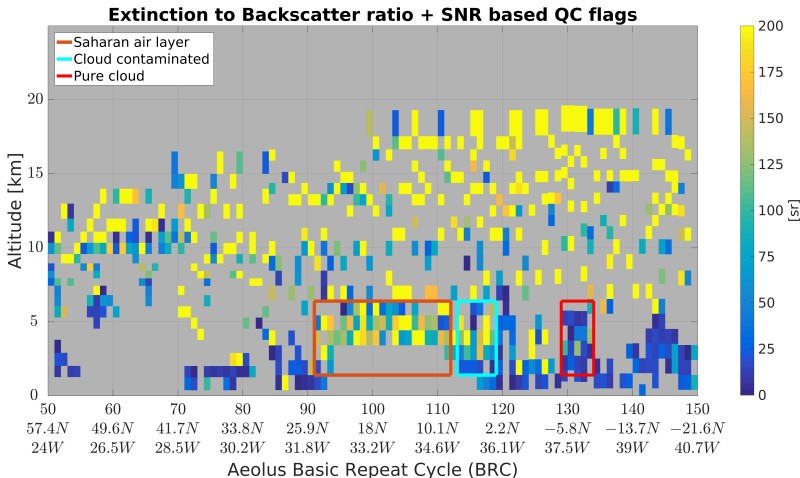

**Figure 15.** SCA mid-bin co-polar Extinction to Backscatter ratio. The red box is the selection for the averaging of the co-polar lidar ratio of the dust plume

coefficient is valid if the Rayleigh SNR in this bin is larger than 90. This allows for the rejection of the bins with low signal, for which background noise is large.

The lidar ratio depends on both extinction and backscatter values and the lidar ratio is valid where both extinction and backscatter are valid. The L2A valid lidar ratio is shown in Fig. 15. Numerous valid bins within the dust plume (orange box) show homogeneous high lidar ratios in the mid-troposphere from altitudes $\approx 2.5$ km to $\approx 5.5$ km. The plume appears to be clearly visible in the data with reasonable contrast despite the background noise (light green and yellow color code). The median lidar ratio within the red box is 130 sr. A ratio of the same order was computed for dust particles emitted from identical Saharan source on 30 June and observed by Aeolus above Cap-Verde (Ehlers et al., 2021a). Because ALADIN only detects the co-polar component of the backscattered light, the backscatter coefficient is actually a co-polar backscatter coefficient. Therefore, in the presence of depolarizing particles, we obtain a co-polar lidar ratio which is larger than what would be expected for an unpolarized observation of the lidar ratio. Earlier on the same day (19 June 2021), CALIPSO flew over the North East Atlantic Ocean from 04:07 UTC and captured the massive Saharan Air layer as shown in Fig. 16. The depolarization ratio at 532 nm reaches up to 0.5 within the dust layer (the large feature on the right of the figure, between 0 and 6 km altitude). This confirms the high concentration of depolarizing particles.

Following Wandinger et al. (2015), the co-polar lidar ratio in circular polarization could then be scaled with a factor $S_{co} = S * (1 + \frac{2\delta_{lin}}{1-\delta_{lin}})$ and compared to other lidar measurements. Mona et al. (2012) report lidar ratios of 40 to 80 sr for mineral dust, with a most likely value around 55 sr at 355 nm, while Nisantzi et al. (2015) report a mean value of $53 \pm 3$ sr at 532 nm for Saharan dust. We obtain values between 80 and 120 sr; from the CALIPSO data in Fig. 16, we estimate the depolarization ratio at 532 nm to be $\approx 0.26$ within the dust plume. The scaling factor to compare Aeolus lidar ratio to a "total lidar ratio" would be $\approx 1.7$, which would scale back our measurements to somewhere between 47 and 70 sr.

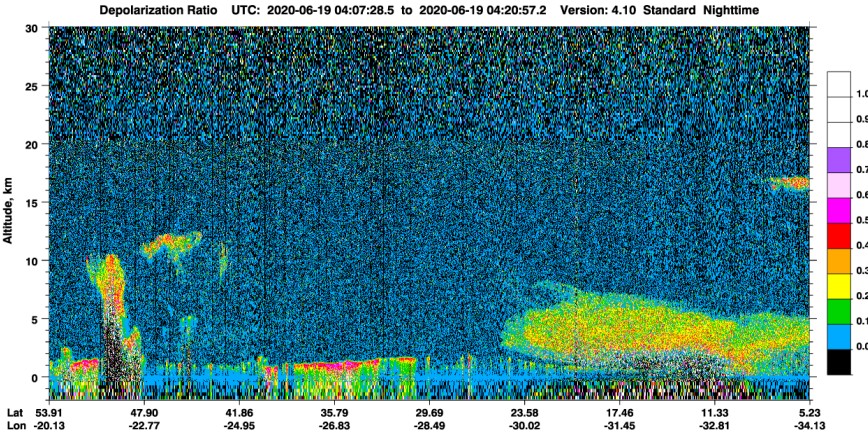

**Figure 16.** CALIPSO overpass above North East Atlantic ocean - depolarization ratio product (CALIPSO, 2021)

## 5 Conclusions

This article gives an overview of the two main algorithms used in the Aeolus mission to derive the Level 2A product, i.e. the aerosol optical properties product. We presented the adapted calibration procedure that was made necessary by the sensitivity of the instrument to thermal strain of the primary telescope mirror. We also analyzed the sensitivity of the extinction retrieval of the SCA in detail so that users can better understand its limitations. The capacity of ALADIN to measure independently backscatter and extinction is also demonstrated on a specific case of Saharan dust transport, where the retrieved lidar ratio is comparable to values published elsewhere.

The L2A product is still under development and new algorithms, better designed to cope with the high noise level of the instrument, are being developed. We can cite for instance, the work of Ehlers et al. (2021b) or the adaptation of EarthCARE algorithms to Aeolus. Yet, we showed the ability of Aeolus to provide valuable information thanks to its HSRL design and despite the high noise and absence of depolarizing channel.

The backscatter measurement is less noisy and the extinction measurement can be made more reliable by using the SCA "mid-bin" algorithm. We recommend that users interested in extinction and lidar ratio read this data set rather than the "normal bin" SCA. The lidar ratio derived directly from Aeolus observations can help discriminating between clouds and various types of aerosols. As ALADIN performs its measurements only along one polarization direction, it will observe a co-polar lidar ratio, and depolarizing targets will appear to have a large lidar ratio. In order to compare Aeolus lidar ratio to other measurement it is important to account for the depolarization ratio, which must be observed by other means.

The Aeolus optical properties product started to be used in validation studies (e.g. Baars et al. (2021)) which showed it could provide valuable data. Finally, assimilation of Aeolus backscatter coefficient into atmospheric chemistry and transport models is being studied and first results are encouraging (Letertre-Danczak et al., 2021).

*Data availability.* Data is available to registered users from the Aeolus Data Dissemination Facility http://aeolus-ds.eo.esa.int/oads/access/

. L2A data access is opened to the public since 12 May 2021.

*Author contributions.* The CNRM contributors are responsible for the code developments and maintenance. Frithjof Ehlers provided analyses
of the L2A results and contributed to the improvement of algorithms. Dorit Huber is responsible for the coding of the operational processor
deployed at ESA. The paper was written collegially by CNRM authors and reviewed by Dorit Huber and Frithjof Ehlers.

*Competing interests.* The authors declare no competing interests.

*Disclaimer.* The presented work includes preliminary data (not fully calibrated/validated and not yet publicly released) of the Aeolus mission
that is part of the European Space Ageny (ESA) Earth Explorer Programme. This includes wind products from before the public data release
in May 2020 and/or aerosol and cloud products, from before their public release in May 2021. The preliminary Aeolus wind products will
be reprocessed during 2020 and 2021, which will include in particular a significant L2B product wind bias reduction and improved L2A
radiometric calibration. The processor development, improvement and product reprocessing preparation are performed by the Aeolus DISC
(Data, Innovation and Science Cluster), which involves DLR, DoRIT, ECMWF, KNMI, CNRS, S&T, ABB and Serco, in close cooperation
with the Aeolus PDGS (Payload Data Ground Segment). The analysis has been performed in the frame of the Aeolus DISC.

*Acknowledgements.* The L2A has been developed over more than 15 years by numerous contributor (Pierre Flamant, Marie-Laure Denneulin,
Mathieu Olivier, Vincent Lever, Pauline Martinet, Hugo Stieglitz ...). Most of the development work for the L2A processor was carried out
in the frame of contracts from the European Space Agency, but also benefited from the financial support of the French space agency CNES.
We acknowledge the use of imagery from the NASA Worldview application (https://worldview.earthdata.nasa.gov/), part of the NASA Earth
Observing System Data and Information System (EOSDIS). We also used data from the NASA/CNES mission CALIPSO and the European
Commission Sentinel program.

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
