# Peer review of "Aeolus L2A Aerosol Optical Properties Product: Standard Correct Algorithm and Mie Correct Algorithm"

_Atmospheric Measurement Techniques, 2021_

## Author Comment (AC1)

Atmos. Meas. Tech. Discuss., referee comment RC1
https://doi.org/10.5194/amt-2021-181-RC1, 2021 © Author(s) 2021. This work is distributed under the Creative Commons Attribution 4.0 License.

Comment on amt-2021-181

Anonymous Referee #1

Referee comment on "Aeolus L2A Aerosol Optical Properties Product: Standard Correct Algorithm and Mie Correct Algorithm" by Thomas Flament et al., Atmos. Meas. Tech. Discuss., https://doi.org/10.5194/amt-2021-181-RC1, 2021

General comments
The manuscript under review describes the standard correct and Mie correct retrieval algorithms as part of the Aeolus L2A data products. It is a high-level summary of those algorithms, meant to introduce the most important aspects and to point readers to the more technical algorithm theoretical basis document. The text meets this objective successfully. The different algorithms are described with adequate detail to get a good picture of their inner workings. Ample evidence is given to justify algorithm decisions and retrieval behavior is characterized using simulated data and case studies using data measured in orbit. The manuscript is organized in a logical manner. There are some minor areas where the text could be clarified, and arguments made stronger. These are noted in the specific comments below.
The topic of the paper is relevant and important for the scientific community to understand the Aeolus L2A data products. It is accessible to a broad audience. The title is appropriate, and the amount of information presented is appropriate for the intentions of the manuscript. The abstract also contains sufficient detail, with only a minor amount of clarification requested. The conclusion section seems a bit too concise without much detail, but all relevant details already exist in the main text. Altogether, this manuscript is well within the scope of AMT and will be welcome asset to the scientific community. My specific comments below are to encourage more clarity in some areas, to make the language more precise, and to bolster a few arguments. These comments could easily be addressed with a minor revision.

Specific comments
Abstract: (optional recommendation) It would be helpful it the abstract mention that the standard correct algorithm and Mie correct algorithm are described in detail. That would help researchers know that this information is included based on the abstract alone.

We added the name of the SCA and MCA to the abstract.

Abstract, line 5: "the theoretical basis is the same as Flamant et al. (2008)". Elsewhere the manuscript says that the processor has "substantially evolved" since Flamant et al. (2008) which seems at odds with this statement.

We reformulated the later text to say that the software evolved while keeping the same "core principles"

Line 94 suggests that "most of the theoretical basis is consistent with Flamant et al. (2008). Could these statements be made consistent?On a related note, the paper mentions "the processor" numerous times. Could some description of what this processor is be given in the main text? Is it meant to describe the code for the entire collection of L2A algorithms? Or just the algorithms

themselves? It seems like the terms "theoretical basis", "algorithms", and "processor" are used synonymously. Should they be?

We acknowledge that we sometimes mix up the term in our daily use and that was reflected in this paper. We trier to clarify the use:
the algorithm is used for the actual equations as described in the theoretical basis document, the processor is the software implementing the algorithms.

Abstract, line 5: The abstract is the only place in the manuscript where the version number (3.12) and baseline number are mentioned. This should probably be described in the main text as well. In fact, it would be helpful to describe what is meant by baseline in terms of the data production strategy for the mission.

We added the following paragraph at the beginning of section 2.2 "L2A product overview".

As the processor evolves, its version number is incremented. This paper describes the processor version 3.12. The processor are then picked up by ESA and integrated to the processing facility. The configurations of this facility are labelled as baselines: each change of processor or major change in processors configurations would trigger an increment in the baseline number. This paper uses the L2A configuration for Baseline 12, which came into Near Real Time production at the beginning of 2021 and was released to the public on 12 May 2021.

Lines 124-125: What are the units of calibration constants K and C? Is this something that should be added to the text?
The C coefficients are unit-less.

The K coefficients are used to scale numerical signals, in "Least Significant Bit" or unit-less to the attenuated backscatters and the laser energy. According to equations 1 and 2, the K coefficients would have units of $m.sr.J^{-1}$.

Line 212: The phrase "full observation file" is a bit confusing, because an there are multiple "observations" (as defined by lines 70-71) in the file. Does omitting the word "file" and just using the phrase "full observation" make the statement accurate?

We replaced "full observation file" by "orbit file", and added the following paragraph to explain what is meant:
"Data is downlinked from the satellite at two ground stations, in Svalbard and at Troll, Antarctica. This is reflected in the product files that are cut at the downlink time. Most of the files are full orbits, e.g. from Svalbard to Svalbard, but some file only cover half orbits, e.g. from Svalbard to Troll while others files cover more than one orbit. These are indiscriminately and improperly called "orbit" files within the Aeolus DISC consortium."

Line 228: How does Figure 6 hint that the extinction is underestimated? If the fixed lidar ratio is supposedly too low and the lidar ratio in Fig 12 is biased high, then how do we know the extinction is underestimated? With figure 6 alone it is difficult to make the argument that the dust extinction is underestimated. Some more detail or a stronger argument should be given here.
The underestimation only comes from the fixed lidar ratio. We added: "(see Fig. 15 for comparison with the SCA lidar ratio)"

Figure 6: It would be helpful to draw a box or otherwise point out the dust plume. It is hard to see otherwise without some sort of annotation.
We wrote "The dust plume is located around BRC #100 at altitudes below 5 km" in the legend.

Line 253: "We will see that the original algorithm..." What is meant by the original algorithm? It is unclear what is being discussed here.
We replaced "original algorithm" with "SCA".

Line 258: "The ATBD describes in detail how we can access the extinction..." Doesn't this manuscript describe how extinction is retrieved? This sentence makes it unclear if the extinction being discussed in this section is from the retrievals already introduced in the paper or if there is some other extinction retrieval described in the ATBD that needs to be understood before reading this section. Clarification is requested.

We extended the sentence:
"The ATBD describes in detail how we can access the extinction within bin i from the available observations. The following section intends to give a physical understanding of the SCA extinction retrieval and its limitations."

Lines 272-273: "This choice of thresholding has been largely discussed." It is not clear how this sentence adds to the justification for using the threshold. Does this just mean to say that the choice was considered carefully?
We meant that there was still a debate whether it is the right approach. We removed the sentence not to confuse the reader.

Lines 275-276: "...that the SCA extinction...lacks sensitivity". What is meant by sensitivity here? Is it the ability to observe weak features? Some more details on the statement would better help understand the limitations of the retrieval.
We rephrased this sentence. We changed "…. lacks sensitivty" by "is unable to detect extinction in bins with an optical depth lower than the cumulated optical depth above."

Equation 18. Some details about this equation should be added, for instance, what are the definitions of the variables?
We better described what is discussed in Eq. 18 and 19.
"where $\sigma_{Lp,i}$ is the standard deviation on particles optical depth in bin i, and $ex_i$ is the error added by the observation $X_i$ on top of the actual value $X_i$, modelled as $X_i = X_i * (1 + ex_i)$"

Line 285: "The loss of vertical resolution is compensated by a substantial gain in errors" Does a "substantial gain in errors" mean that there is a substantial improvement in errors? The word "gain" is ambiguous because it could mean that there is an increase in errors, in which case both vertical resolution and accuracy are lost. "Improvement" is clearer.
We replaced gain by improvement.

Figure 8 and line 288. The text argues that the SCA mid-bin retrieval is the best choice for extinction because it would eliminate the stripes in the right-hand panel of figure 8. It would be useful to show the same example for Figure 8, but with the mid-bin retrieval to demonstrate how it improves the striping.
The mid-bin retrieval doesn't apply the threshold to keep extinction positive, there are not stripes. We added the mid-bin retrieval to figure 8.

Line 342: What quality flags have been applied? Is it just the SNR thresholds discussed in the paragraph? If there are more quality flags applied then it would be helpful, especially for data users, to state which flags are used.
We rephrased as follows:
"The SCA lidar ratio in Fig. 15 is read from the mid-bin product and quality flags have been applied. According to these flags, the backscatter coefficient retrieval is considered as valid in a

specific bin if the Mie SNR is larger than 40 and the extinction coefficient is valid if the Rayleigh SNR in this bin is larger than 90. This allows for the rejection of the bins with low signal, for which background noise is large."

Lines 352-353: The text states that the lidar ratio appears higher than other lidar measurements due to because only the co-polar channel is measured by Aeolus. How much higher is it reasonable to expect the lidar ratio to be due to this? Is it meaningful to estimate the depolarization of the dust plume and the subsequent expected overestimation of lidar ratio? Is that outside the scope of this manuscript? It would instill more confidence in the retrieval to know that the overestimation of lidar ratio in this example is consistent with what is expected due to the missing cross-polar channel rather than some other retrieval artifact or calibration bias.
The explanation on the depolarization and its impact on backscatter measurement has been reworked and expanded. We also provided values for the conversion of the co-polar lidar ratio derived from Wandinger et al. (2015).

Line 366: What is the "basic cloud classifier"? Is that part of the L2A data products? The remainder of this paragraph relies on determinations of the cloud classifier and cloud mask, but sparse details are given on how it works. Adding a reference to more information about this classifier would help readers understand its limitations in this analysis.
This paragraph was removed along with the last figure presenting the AUX_MET data.

Section 5. The conclusions section is missing a summary of the algorithms discussed in the paper. Maybe this section is meant to be concise, but it seems incomplete. It would be more informative if it restated the main algorithms discussed and quantities retrieved. Even better, it would be helpful to state which of these retrievals are recommended for users. All this is in the main body of the text, so my comment here is an optional suggestion for the authors.
We added a short summary at the beginning of the conclusion and added references to future algorithms, assimilation work and a validation study.

Technical corrections
Line 32: Remove unnecessary word, "In", ..."CALIPSO In is an older lidar mission…" Done
Line 35: Reference should be Omar et al., 2009 instead of Ali H. et al, 2009 Corrected
Line 65: Should say, "depicted in Figs. 1 and 2." Corrected
Equation 2: Subscripts for S and K should be mie instead of rie. Corrected
Line 139: Extra parentheses at beginning of Dabas reference. Removed
Equations 7 and 8: Is the subscript "i" missing or was it intentionally ommitted? It appeared in equation 2. It was ommited to lighten the equation, the subscript was added for clarity.
Line 221: The subscript for molecular extinction in the equation for molecular attenuation is incorrectly given as "p" instead of "m". Corrected
Line 226: Should be "This yields..." rather than "This yield…" Corrected
Line 270: The word "is" is unnecessary..."the SCA extinction is only yields…" Removed
Line 276: Unnecessary parentheses at the end of this sentence. Removed

---

## Author Comment (AC2)

Atmos. Meas. Tech. Discuss., referee comment RC2
https://doi.org/10.5194/amt-2021-181-RC2, 2021

[Figure]

**Comment on amt-2021-181**

Anonymous Referee #2

Referee comment on "Aeolus L2A Aerosol Optical Properties Product: Standard Correct Algorithm and Mie Correct Algorithm" by Thomas Flament et al., Atmos. Meas. Tech. Discuss., https://doi.org/10.5194/amt-2021-181-RC2, 2021

The paper describes in a short and concise way the algorithms for the retrievals of the aerosol and cloud product from Aeolus. It is thus very informative and valuable for the scientific community and therefore, in principle, well suited for publication in AMT.

However, I have some major concerns which need to be addressed before the paper can be published. The authors have done great work in developing the algorithms for Aeolus and updating calibration schemes, but the current presentation style of the paper needs to be clearly improved otherwise it is not understandable and thus publishable. Therefore, most of my comments are with respect to that topic.

**General remarks:**

The paper tries to make a compromise between extended algorithm description and concise information. This, however, was not successful all time. Especially, a lot of "Aeolus internal" language is used, which is not understandable for an external readers. Some examples are given below, but please check that everything is explained and clearly references so that a person with no access to the internal ESA pages can understand everything.

Furthermore, the main ESA documents which the authors reference on (e.g. the ATBD) should be made available in a sustainable way. Currently they are published on an ESA webpage, but who knows if this is still the case in 1 or 2 years. Thus, please either put this important information on a repository where you can obtain a DOI (e.g. zendo) OR submit it as supplementary material.

Furthermore, while progressing with the reading the paper, the language style gets more and more sloppy and clearly needs to be improved (to be honest, one has the feeling it was submitted before it was really finalized, i.e. some sections are still in a draft-stage). E.g., the current conclusion is not sufficient and not appropriate for a journal like AMT. Also, the language itself is partly not

scientific and still a lot of typos exist. Thus, this should be improved during the revision or language editing should be made by Copernicus.

In general section 4.1. and 4.2 has to be overworked. The explanations are partly insufficient and one has to guess many times what is meant…..

References: The references given in the introduction and the paper are not up to date (one has the feeling the list is 2-3 years old). Meanwhile, some papers have been published, also dedicated to aerosol and cloud products, and should be mentioned. Some examples are given in the specific comments.

Figures: Please explain each Figure you use and what can be seen in this Figure. Currently, very often you draw conclusion from a Figure, but for an external reader it is not comprehensible/understandable because it is not sufficiently explained what is shown in the Figures. In principle, you need to explain each Figure in the text, and additional as a self-standing description in the caption. So that one could understand the Figure from reading the caption only, but also from reading the text only. Furthermore, please check which Figures you really need to make you message: "Illustrations should only be shown if they are necessary for the understanding of the paper, not because they have been created. "

**Specific comments:**

Line 26: "molecular photons" do not exist, I guess you mean photons backscatter by molecules.

Corrected

Line 32: delete IN (after Calipso)

Done

Introduction in general. Please review the current status of Aeolus (space lidar) related literature and add the important most recent references.

Lind 35: Wrong naming in reference, it is OMAR et al and not Ali et al.

Corrected

line 41: add "and" before nature

Done

Line 52: Sustainable source for Flamant 2021 needed.
The document was added to a specific documentation page for the L2A on the ESA Aeolus page. A request for a DOI was addressed to ESA.

Line 66: Add "Fig." before 1 and 2. (do you need these figures?) added "Fig.". We thought these figures would be useful to underline Aeolus particular slant looking geometry and the range bins.

Line 79: "the top-most Rayleigh bin that must be above the top-most Mie bins" Please explain why and/or give reference.

This is a built-in technical constraint of the hardware, which is now explained in the text. References are only ESA internal documents.

84: "The shape of the optical filters is drawn in Fig. 3.". Language! What is shown in Figure 3 are the transmission curves for the different channels/filters.

Corrected

86: " The figure shows that the Mie peak in the spectrum is significantly filtered out by the dual Fabry-Perot as it stands half-way between the peak transmissions of Fabry-Perot A and B, where the sum of the two transmissions reaches a local minimum." I assume that you reference now to the right panel of Figure 3. I took me quite some while to understand what you have written. You need to rewrite this paragraph adding more explanation. Please guide the reader to what is the "Mie peak". Explain all abbreviations in the Figures (e.g. what is TA and TB?) and use the line colour and style when referring to a specific curve to help the reader understanding.

The explanation on this figure has been expanded and the figure legends changed.

90: "Overall, the efficiency of the Rayleigh detection chain for the photons backscattered by particles is about 50% of what it is for molecular photons, while it is 130% through the Mie detection chain." The current phrasing is very, very hard to understand. Please try to rephrase to make it clearer for the reader.

We tried to rephrase these paragraphs to make them easily understandable. We also give a figure with the actual C coefficients to illustrate this paragraph.

Figure 3: y-Axes caption on the right column missing. X-axis caption: What is 0? This is never explained. I now it is the difference to the emission frequency but you need to state this. Thus, it is not the frequency but the frequency difference/shift.

The frequency offset is now explained in the text is corrected on the figure. The y-axes are spectral power distribution, in arbitrary units.

Line 99: L1B never explained. What is this, any reference? I think all, the L1B and L2A ATBD and product description documents need to be published with a DOI from the current versions.

We added a short explanation of what the L1B input data that we use. Having a DOI would be good indeed. We asked ESA to consider this in the future. The L2A documents were also provided on a specific page, other than the initially referenced "announcement of opportunity" page.

106: L1B derived scattering ratio is defined by the physical quantities. But how can this be done at L1B level?

See response to the above comment.

107: Which pre-defined lidar ratio is used? - please state this here.

The backscatter-to-extinction ratio used in the L2A was added.

112: Please explain SNR and also please explain what "high" means.

We added an explanation:

"...with large enough Signal to Noise Ratio (SNR) within one BRC before applying the SCA. The threshold is set at a SNR of 3.5 in Baseline 12."

112: Is the group product really limited to one BRC? I thought features are grouped on the basis if "measurements" independent of the BRC.

Groups are processed in each BRC separately. It comes with the limitation that no group can be larger than one BRC and no group can spread across the border between two BRCs.

122: $S\_rie$ --> $S\_Mie$

Corrected

130: add "and" after the formula

Done

133ff: As C1 to C4 are fundamental coefficients, the short explanation is not sufficient to me (as the reference given is a zip file only and no peer-reviewed document). Especially the adhoc calibration procedure (line 137) needs a short and concise explanation here. Also the uncertainties related to that should be briefly discussed: E.g. how good are your C1 to C4 determined and what happens if the calibration fails.

A short description of how the coefficients are derived is now provided.

Errors on C and K coefficients are not accounted for in our error estimation. This is a flaw, as our models are not as precise as it was hoped before launch. We added the following discussion on errors:

"Currently, the error on the calibration coefficients is not accounted for in the error estimates that are derived below. The models used in the calibration are not able to describe imperfections of the instrument that were discovered in flight. In particular, the $C_3$ coefficient is derived from simulated transmission curves, and comparison of transmissions and simulations on the internal path show discrepancies of up to 20 %. The actual transmission of the Fizeau interferometer for a light beam backscattered from the atmospheric (the "atmospheric path") is difficult to calibrate. This improved calibration and the inclusion of calibration errors into the error estimates could be addressed in future versions."

144: I guess you invert not only Eq. 2. but also Eq. 1? At least id did not understand how to achieve 7 and 8 by using only Eq. 2. Please also write inverted "to".....

We changed the sentence to:

"In the first operation of the SCA, for each bin i, equations (1) and (2) are inverted to:"

Eq. 7 and 8: $C_3$ subscript for "3" missing

corrected

155: maybe rephrase to "the assumption is made that within the first bin no particles exist".

We changed the sentence to:

"the assumption is made that there are no particles within the first bin and hence ..."

159: Eq 12: $\beta\_m$ needs subscript sim as well?

Yes, we added the subscript

Line 167: What is x – it's not explained. In general formula 14 is hard to understand. Do the brackets behind H^-1 correspond to x? That is unclear for me. Probably it would be better to explain this formula in two steps or you simply do not use "H".

We need to use $H^{-1}$ because there is no formula to describe this function. In the software, H is being inverted numerically. "x" is just a variable to explain what function H is (In France it is rather usual to call "x" any scalar variable, that we can use to describe a function, sorry if that was not clear). We rewrote this part as follows.

"Where H is the function:    $x \rightarrow H(x) = \dfrac{1 - e^{-x}}{x}$    "

Figure 4 is never referenced. I think this should be done as it is very interesting.

Figure 4 is now referenced in the text.

169: For my own interest: Did you ever estimate to what extent possible extinction above the first bin could influence the SCA? Is the normalization procedure probably more prone to produce high errors than a possible extinction above the first bin?

The normalization is here to get rid of this effect. As you noticed, however, the normalization is very prone to errors because of the low level of signal in the uppermost bin. We did not consider to change this part of the algorithm.

The future Maximum Likelihood Estimation algorithm will allow to determine the optical depth between the satellite and the first bin. This is something that we will be able to study more in depth in the future.

180: Please explain "M1". E.g.: "...the primary mirror (called M1)". But probably this is not needed. Do you have by the way any reference for that statement? E.g. during last ILRC, a lot of Aeolus presentations were made:

https://www.epj-conferences.org/articles/epjconf/abs/2020/13/contents/contents.html or even that manuscript: https://amt.copernicus.org/preprints/amt-2021-171/   ?

We added a reference to Weiler et al. 2021.

188: Does it impact the retrieval or the calibration?

We rephrased to:

"This change in the instrument characteristics would require a constant recalibration. In the absence of such a correction, both the winds (Weiler et al., 2021) and the aerosol optical properties retrieval would be impacted."

Line 190: Do you have any reference how "clear sky" is defined? I.e. which measures do you apply?

We added a description of the "clear sky" selection method that we apply.

Line 196/197: What is meant with step 2 in Figure 5? Is is not clear for me. More explanation is needed here.

We replaced "step 2" by "the second row".

Fig. 5: I think Fig 5, left is not useful to understand the paper. If you want to keep it, it must be enlarged, colour scale must be changed and much more explained. But from my point of view, Fig. 4 and Fig 5. (right) are sufficient. Nevertheless, it is surprising to see a big difference in Fig 5. left for the orbit averaged k and the M1 fitted K, while in the distributions on the right side it is obviously not the case. Can you explain? Virtually you have much more values around or above 4 for the orbit averaged K but this is not seen in the distribution (right panels).

We replotted figure 5 with a broader range of values but there is no visible difference between the $2^{nd}$ and $3^{rd}$ distributions. We are still wondering why the "recalibration" option produce such broad distribution of errors.

We would like to keep the left panels as they give a different on the correction. We described them better.

206: please explain what "distribution" you mean. I guess you refer to Fig. 5 right, but you need to explain what is shown there.

We added: "on the right panel".

211: "The fit being made…..": which fit? It is unclear for externals what is meant with all this. Please explain more solid in scientific language.

We rephrased:

"The fit of the signal to the M1 temperatures is made over a full "orbit file", which is at least half an orbit long and most of the time longer, this is enough to guarantee that a high reaching particulate feature in a given region would not bias the fit too much."

216: "L1B derived scattering ratio":  Never explained. What is this?

Eq. 16: What is roh_L1B,I ? Is this the scattering ratio. What is the difference between a real scattering ratio and the L1B derived one. Need to be explained.

We introduced the L1B product in a dedicated section. This was indeed missing.

224: Tm,sat,i-1 is not in Eq.  17. Please correct.

"sat" subscript was added for both Tm and Tp, as it should have been.

228: Which lidar ratio did you use and why? This is an essential information.

We added:

"… under-estimated due to the fixed lidar ratio of ~14 (1/0.07) used by the MCA (see Fig ...)"

228:  How one can see a dust plume? Please indicate in Fig. 6 and maybe also put geophysical coordinates to Fig. 6 (as for the Calipso image or Fig.12).

We added geographical coordinates to figure 6.

Fig. 6: The size of this Figure is good as well as the color scale. Some more explanation is needed in the text: what are the white areas, why do the top of the profiles changes. Where has this curtain been made,  etc……

On top of adding the coordinates along the curtain, we added the following paragraph:

As explained in \ref{aladin_and_aerosol}, the Range Bin Setting (RBS) is changing along the orbit in order to find a compromise between the highest sampled altitude and the resolution. This is visible in the big steps in the maximum altitude of the profile. Smaller steps are due to the terrain following capability( e.g. towards the end of the the orbit), that shifts the RBS to reduce the amount of data acquired below the ground and extend the profile higher up. Some parts of the profiles, especially in the lower atmosphere are not processed (white pixels). This happens when the measured Mie signal becomes negative, often below thick clouds.

231-239: I am puzzled how I should deal with this information. So the ICA is kept in the data for historical reasons. But no development have been made. What does it mean? Shall I neglect the ICA? Same for the group product. A clear statement would be desirable.  Or do not describe the ICA and group at all (maybe only in the introduction) as it is not used in your analysis. At least, in the current form it might be more confusing for the reader than providing valuable information.

This paragraph was removed and only mention these parts of the product in the introduction.

245: As far as I know, ALADIN is not linearly but circular polarized….thus there is also no parallel direction….

The explanation on the polarisation has been rewritten to give more details. This is not well known by the community and was pointed by the three reviewers, so we tried to make things clear.

245-247: Please put reference here. E.g.  Ansmann  et al., 2007;  Flamant  et al., 2008, or Baars et al. (2021).

This paragraph has been further developed and we added a reference to the ESA Aeolus "Science Report":

"Designed as a wind lidar, ALADIN was not initially aimed at observing aerosol optical properties in detail. Under these requirements, it was not fitted with the ability to measure depolarization. The UV laser beam is linearly polarized at the laser output. It goes through a quarter-wave plate (see Fig. 4.13 in (Science Report)) before being routed towards the telescope and is thus transmitted towards the atmosphere with a circular polarization. On the way back, backscattered light goes again through the quarter-wave plate. The circularly polarized light that was transmitted might come back elliptically polarized in the case it was backscattered by depolarizing targets. After going through the quarter-wave plate is a mix of linearly polarized light, along the same direction as the transmitted light (co-polar) or along the perpendicular direction (cross-polar). The beam then reaches a polarized beam splitter. The co-polar light is routed towards the interferometers, while the cross-polar light is routed back towards the laser and is lost for the analysis. This means that, in order to compare Aeolus observations of backscatter coefficient and lidar ratio to other instruments, only the co-polar component must be considered."

255: Any reference for that statement that signals are weaker than expected as before launch? e.g.: Reitebuch,2020, ILRC or even directly in this special issue?

We added a reference to Reitebuch 2020.

268: "See Fig. 7.": More explanation for Figure 7 needed. E.g. which plot in Fig. 7 is meant, what is shown there, etc. Just to refer to a Figure without any explanation what is shown there is not sufficient. Furthermore, many things shown in Fig. 7 are never discussed, e.g. backscatter *30sr….

We changed see Fig. 7 into:

" This is illustrated in Fig.7, where a large extinction is found in the second bin. This produces a large attenuation on the expected molecular signal (red dashed line) which never becomes larger than observed molecular signal (yellow line)."

We also explained the "backscatter*30 sr" line:

"As an indication of the presence of particles, we also show the SCA backscatter scaled by an arbitrary lidar ratio (middle panel, red line). It shows that the extinction of the SCA (blue line) is detected one bin below the actual particle feature. The MCA extinction is quantitatively wrong because of the fixed lidar ratio, but is detected in the correct bin."

274: Same as above but for Figure 8. E.g., Fig. 8, left is never referenced. And it is never explained what is seen there in general. Moreover, the panels should be enlarged to page width and been put over each other.

Fig. 8 was redrawn. We put panel on top of each other and added the mid-bin extinction for comparison.

Eq. 18: most of the quantities shown in this equation are not explained, thus one cannot follow the argumentation and understand the formula.

This is better explained now:

"where $\sigma_{L_{p,i}}$ is the standard deviation on particles optical depth in bin i, and $e_{X_{i}}$ is the error added by the observation $X_i$ on top of the actual value $\overline X_i$, modeled as $X_i = \overline X_i * (1 + e_{X_i})$."

Fig. 19: Same comment as for Eq. 18. Please discuss the equations or review if you really need to show them. I could not follow any of the argumentation from 277ff.

We added:

"This estimated standard deviation $\sigma_{L_{p,i}+L_{p,i+1}}$ is no longer linked to $e_{X_1}$, the error on $X_1$, but only to the error in the two bins that are combined to obtain the "mid-bin" value."

285: If you "lose" vertical resolution but also gain errors, why to use this method? I guess you mean lose resolution and decrease errors?

We replaced "gain error" by "improvement in error"

296: two times "presented", delete one of it.

Done

299: E2S never explained, please do so when introducing the end-to-end simulation. Furthermore, what is the difference of the 20 simulation? It is not written here. If they are produced from the same input scene they should deliver the same results unless you alter some parameter. Which ones? what was simulated?

We added "(E2S)" when we introduced the End-to-End Simulator, earlier in the section.

To explain our approach, we added: " The noise generated in each simulation is different and this allows us to estimate the impact of noise separately from the potential biases of the algorithms"

301: "Most of the time, the backscatter and extinction coefficients are correctly derived" how is this seen? What are you looking at?

See comment about line 316

Figure 10 and discussion: The current Figure is hard to read. It is 16 panels with 6 curves each. Do you really need all panels to make your statement? Maybe show only the most important. Please also explain all abbreviations and formula symbols. Why do you use log-scale for the backscatter and extinction values?

See comment about line 316

Maybe you could start introducing the reader to these kind of Figures by grabbing one BRC and first explain in detail what is shown. Afterwards show the other BRC's and do your interpretations. But currently you ask too much from the reader to understand what you see in these plots.

See comment about line 316

302: "i.e. errors lie within the range of atmospheric heterogeneity" – how is this heterogeneity determined. It is currently a statement without proof.

See comment about line 316

303: "backscatter coefficients are also mostly correct" what does this mean, where it can be seen?

See comment about line 316

304: "The average of the 20 simulation overlap the expected values with a low dispersion meaning that one realization should be enough to characterize the atmospheric optical properties." I do not understand this sentence as I do not know to what you are referring to.

See comment about line 316

307: "In practice the vertical resolution of the bins is seldom below 500 m" has it ever be explained that the range-bin setting can be changed and is changed along one orbit? This is an important information….

We completed the corresponding paragraph in section 2.1. It now reads:

"Most of the time, the backscatter and extinction coefficients are correctly derived, which can be seen where the red line (20-run average retrieval for a given BRC) is close to the black line (E2S input averaged over the BRC). Error estimates for the extinction (brown range around the red line) lie within the range of atmospheric heterogeneity (thin black line, derived as the standard deviation of the E2S input over a given BRC). Backscatter coefficients are also mostly correct, although with a slight low bias, e.g. in BRC 5 between 14 and 10 km altitude (compare the red line to the input, black line). The average of the 20 simulation overlap the expected values with a low dispersion (brown-shaded area) meaning that one realization should be enough to characterize the atmospheric optical properties. "

299-309: The paragraph should be overworked in general. For me it was hard to understand to what the authors refer to when making a statement.

We tried to better point at what to look at in the figure. The paragraph now reads:

"In order to study the sensitivity of the L2A product to noise, 20 independent E2S simulations are run from the same input scene. The noise generated in each simulation is different and, looking at the average retrieval and the standard deviation around it, we can estimate the impact of noise separately from the potential biases of the algorithms. Figure \ref{SCA_vs_E2S} presents how the backscatter and extinction coefficients derived from the SCA mid-bin algorithm compare with the E2S inputs.

Most of the time, the backscatter and extinction coefficients are correctly derived, which can be seen where the red line (20-run average retrieval for a given BRC) is close to the black line (E2S input averaged over the BRC). Error estimates for the extinction (brown range around the red line) lie within the range of atmospheric heterogeneity (thin black line, derived as the standard deviation of the E2S input over a given BRC). Backscatter coefficients are also mostly correct, although with a slight low bias, e.g. in BRC 5 between 14 and 10 km altitude (compare the red line to the input, black line). The average of the 20 simulation overlap the expected values with a low dispersion (brown-shaded area) meaning that one realization should be enough to characterize the atmospheric optical properties. "

316: "The estimated errors are also too low and do not cover the expected values": How can I see that in the plot? Unclear for me.

The whole discussion on this figure has been reworked, with explanations on the figure itself and what to look at in the figure. We kept the figure as it was and we hope that the text is now enough to guide the reader through it.

319: "In this example," which one?

We changed to "In Fig. 10"

320: It has been never explained what a "useful" signal is…..

The input data from the L1B is now introduced in a dedicated section.

322: "bias is then propagated up to the calculation of the backscatter" what does it mean: propagated up to? Please check if you can find a proper peer-reviewed reference for CALIOP, e.g. Winker 2009

The sentence was corrected: "The underestimation of the Mie SNR is then propagated through the derivation to the estimated variances of the backscatter and extinction."

We added Winker, 2009

343: "determined by a threshold on the Mie SNR…" what is the threshold?

We rephrased the paragraph:

"The SCA lidar ratio has been processed from the mid-bin product and SNR based quality check (QC) flags have been applied. The backscatter coefficient retrieval is considered as valid in a specific bin if the Mie SNR is larger than 40 and the extinction coefficient is valid if the Rayleigh SNR in this bin is larger than 90. This allows for the rejection of the bins with low signal, for which background noise is large."

345: "reject low signal bins" please improve phrasing, what are low signal bins, bins with low signal?

We changed the sentence to "This allows to reject bins with low signal, for which background noise is large."

346: "L2A valid lidar ratio" what does valid mean? I guess you mean that you applied the validity flags?

We added the sentence:

"The lidar ratio depends on both extinction and backscatter values and the lidar ratio is valid where both extinction and backscatter are valid. "

Figure 12: What you show is the co-polar lidar ratio, please indicate this in the Figure to avoid confusion. The blue and green frame is hardly seen. Can you use a different color?

We changed the color of the boxes.

347: "high lidar ratio values"…please indicate numbers – it is hard to see from the colors, e.g. 120 - 140 sr. In my opinion you always should state that "only" the co-polar lidar ratio is measured, otherwise readers only looking at the plots may be really confused why the lidar ratio in dust is 2-3 times higher than normal. And you also should state what (co-polar) lidar ratio one would expect in mineral dust. Otherwise the reader is left alone in interpreting if Aeolus L2A data is useful….

We wrote a discussion about the co-polar lidar ratio values and how they compare to the values in the cited literature.

350: You compare apples to peaches: Please state what lidar ratios (numbers) they have been measured (Mona and Nisanzi) and what you would expect for Aeolus taking into account the polar component. There was also a presentation by Wandinger showing that.

We now give values and compares them to the cited articles, using the formulas for circular depolarization ratio from Wandinger, 2015 (Aeolus Workshop):

"Following Wandinger et al. (2015), the co-polar lidar ratio in circular polarization could then be scaled with a factor $S\_co=S*(1 +2\delta\_lin/(1−\delta\_lin))$ and compared to other lidar measurements. Mona et al. (2012) report lidar ratios of 40 to 80 sr for mineral dust, with a most likely value around 55 sr at 355 nm, while Nisantzi et al. (2015) report a mean value of 53±3sr at 532 nm. We obtain values between 80 and 120 sr; using a rough estimation of the depolarization ratio at 0.3 from CALIPSO, the scaling factor would be 1.85. which would scale back our measurements to somewhere between 43 and 65."

353: "A number of studies (Ansmann et al., 2003) have shown that light depolarization ratio of dust and marine particles mixture is significant." I do not understand this sentence.

This sentence was removed.

358: In my opinion, you devalue Aeolus with no need. The lidar ratio is not overestimated taking into account the Aeolus capabilities. Even more, it is absolutely correct when considering the expectations, e.g., made in Wandinger et al. Thus, you might reconsider your statements here.

We rephrased this sentence.

362ff and Figure 14: Is in my opinion not needed. First, it is "only" model data and therefore only an indicator, second it does not provide any additional valuable information. Thus, consider to omit this. If you consider it as very important, than much more explanation is needed.

We removed this figure and the corresponding paragraph.

As written before, the conclusion seem to be unfinished (i.e. still in draft stage) and are not sufficient in the current form. Please revise.

We expanded the conclusion and provided references to others works related to the L2A (new algorithms, assimilation, validation studies )

**References:**

Ansmann, A., Wandinger, U., Le Rille, O., Lajas, D., & Straume, A. G. (2007). Particle backscatter and extinction profiling with the spaceborne high-spectral-resolution Doppler lidar ALADIN: Methodology and simulations. *Applied Optics*, 46(26), 6606– 6622. https://doi.org/10.1364/AO.46.006606

Baars, H., Radenz, M., Floutsi, A. A., Engelmann, R., Althausen, D., Heese, B., et al. (2021). Californian wildfire smoke over Europe: A first example of the aerosol observing capabilities of Aeolus compared to ground-based lidar. *Geophysical Research Letters*, 48, e2020GL092194. https://doi.org/10.1029/2020GL092194

Flamant, P., Cuesta, J., Denneulin, M.-L., Dabas, A., & Huber, D. (2008). ADM-Aeolus retrieval algorithms for aerosol and cloud products. *Tellus A: Dynamic Meteorology and Oceanography*, 60(2), 273– 286. https://doi.org/10.1111/j.1600-0870.2007.00287.x

Reitebuch, O., Lemmerz, C., Lux, O., Marksteiner, U., Rahm, S., Weiler, F., et al. (2020). Initial assessment of the performance of the first Wind Lidar in space on Aeolus. *EPJ Web of Conferences*, 237, 01010. https://doi.org/10.1051/epjconf/202023701010

Wandinger et al., Validation of ADM-Aeolus L2 aerosol and cloud products employing advanced ground-based lidar Measurements (VADAM), ADM-Aeolus Science and CAL/VAL Workshop, 2015.

Weiler, F., Rennie, M., Kanitz, T., Isaksen, L., Checa, E., de Kloe, J., Okunde, N., and Reitebuch, O.: Correction of wind bias for the lidar on-board Aeolus using telescope temperatures, Atmos. Meas. Tech. Discuss. [preprint], https://doi.org/10.5194/amt-2021-171, in review, 2021

Winker, D. M., Vaughan, M. A., Omar, A., Hu, Y., Powell, K. A., Liu, Z., et al. (2009). Overview of the CALIPSO mission and CALIOP data processing algorithms. *Journal of Atmospheric and Oceanic Technology*, 26(11), 2310– 2323. https://doi.org/10.1175/2009jtecha1281.1

---

## Author Comment (AC3)

Atmos. Meas. Tech. Discuss., editor comment EC1 https://doi.org/10.5194/amt-2021-181-EC1, 2021  $\ensuremath{\mathbb{C}}$  Author(s) 2021. This work is distributed under the Creative Commons Attribution 4.0 License.

**Comment on amt-2021-181**

Ad Stoffelen (Editor)

Editor comment on "Aeolus L2A Aerosol Optical Properties Product: Standard Correct Algorithm and Mie Correct Algorithm" by Thomas Flament et al., Atmos. Meas. Tech. Discuss., https://doi.org/10.5194/amt-2021-181-EC1, 2021

Review of "Aeolus L2A Aerosol Optical Properties Product: Standard Correct Algorithm and Mie Correct Algorithm" by T. Flament et al.

\_\_\_\_\_

General Comments

\_\_\_\_\_

This paper gives the impression that the evaluation/validation (and maybe development) of the aerosol/cloud products from Aeolus is still in the early stages, in marked contrast with the aeolus wind retrievals1. The paper presents only one example using real data where the lidar ratio results may be plausible. Almost 3 years after launch, I would have hoped for a more advanced state with respect to the aerosol/cloud product evaluation/validation.

It could well be argued that this paper is premature, however, having said that there are also reasons why this paper is potentially publishable at this time. This paper could serve as a point of reference for the lidar community and to serve as an accessible introduction to the instrument and the existing L2 aerosol/cloud retrieval algorithms. With regards to the later point, to be useful, the presentation of the paper must be improved. I found several areas to be more confusing than illuminating and, at times, the presentation seemed geared more towards "Aeolus insiders" rather than the wider lidar aerosol-cloud community.

The paper was also "thin" on examples using real observations imparting on the reader of the paper no real feel at all for the quality of the data. To this end, the authors should include additional examples, for example, showing:

-profiles and 2-D plots of the Aeolus Attenuated backscatter (both before and after cross-talk correction).

-profiles and 2-D plots of the retrieved extinction and backscatters.

-comparisons of the extinction and backscatter retrieval results for the MCA and SCA algorithm.

The above examples should, ideally, span an appropriate number of representative cases (e.g. cirrus clouds, light and heavy aerosol loadings etc..)

My specific comments follow.

First off, I am puzzled by the use of "correct" in the name of the algorithms refered to by this paper. There are other approaches to inverting HSRL signals to derive extinction and lidar ratio that are mathematically valid. What is special about these algorithms that make them "correct" ? It would be useful to the reader if this point was somehow addressed in this paper.

We added the following discussion in section "high noise and extinction retrieval":

" The SCA is very similar to the classical log-derivative algorithms but the thickness of ALADIN range bins (up to 2 km) mean that the particulate optical thickness (\$\alpha\_p \* \Delta R\$) can be large and the approximation used for the molecular extinction (Eq. 6.34 in \

cite{flamant\_aeolus\_2021}) cannot be used for \$\alpha\_p\$. This is why we later need to inverse function H rather than simply derive the logarithm of the attenuation of the Rayleigh signal. As a side note, this refinement is also the reason why the adjective "correct" was added to the name of the algorithm."

Abstract Line 1:
* * *
"Although ALADIN is optimized ....."

Corrected

Abstract: Line 11:
* * *
The last line is badly worded. I suggest "This is illustrated using Saharan dust aerosol observed in June 2020".

We changed the sentence to the reviewers suggestion.

Page 1: Line 19:

\_\_\_\_\_

I find this short description awkward and not accurate enough. I suggest something like: "Two separate main optical detection channels are implemented on-board ALADIN. They are referred to as the Mie and Rayleigh channels. Both channels detect a mixture of particulate and molecular

scattering. However, the primary task of the so-called Mie channel is to detect the spectrally narrow (FWHM on the order of 10s of MHz) return from atmospheric hydrometers. The Rayleigh channel primarily detects the spectrally broader (FWHM of several GHz) backscatter from atmospheric molecules."

We adopted the reviewer's suggestion.

Page 2: Line 32 ------Delete the "In" directly after the reference to Winker et al. Done

Page 2: Line 33
* * *
2006 was 15 years ago. I think you can delete the "already"...perhaps "previously" was meant.

We removed "already".

Page 2: Line 34

Please be specific. What is "all the available information" ?

We changed to: "... is estimated using information from several wavelengths and a depolarization channel"

Page 2: Line 46.
* * *
This is a very interesting point. Please provide a reference (even if it is only a tech note or report).

We added a reference to an EGU presentation:

Letertre-Danczak, J., Benedetti, A., Vasiljevic, D., Dabas, A., Flament, T., Trapon, D., and Mona, L.: Aerosol Assimilation of lidar data from500Satellite (AEOLUS) and Ground-based (EARLINET) instruments in COMPO-IFS., other, pico, https://doi.org/10.5194/egusphere-egu21-4799, https://meetingorganizer.copernicus.org/EGU21/EGU21-4799.html, 2021

**Page 2: line 53**

\_\_\_\_\_

Please mention how can the general community get access to the updated L2A ATBD.

This is indicated in the reference. We changed the sentence from "can be found on-line" to "can be found on the ESA reference page :https://earth.esa.int/eogateway/catalog/aeolus-l2a-aerosol-cloud-optical-product". We also requested ESA to investigate the possibility to obtain a DOI, which is not ready at the time of answering this review.

**Page 2: Last line**
* * *
"...followed by a conclusion" ==> "...followed by a conclusion section".

**Corrected**

Page 3: Line 68
* * *
I am confused by the reference to the "..previous 24 sec cycle of the burst-mode operation of the laser". Previous to what ? Was this burst mode used early on in the mission ? If so, why was it no longer used ? Or, was it something previously planned but not implemented ?

We clarified this paragraph by saying that some of the parameters foreseen at the time of writing the science report were changed before launch. Among others, the discontinuous burst mode has been replaced by a continuous mode of observation.

Page 3: Line 75

"Fine bins.." ==> "Finer resolution bins..."

We changed "fine" for "thin bins"

Page 3: Lines 83-94 and Fig. 3

\_\_\_\_\_

I found that the discussion of the spectrometers to be very confusing ! Only after reading through the L1 and L2 ATBDs, it became clear that the Rayleigh A and B signals are the result of integrating the images projected on CCD detectors. So for the Rayleigh channels, for each time-height bin two spectrally integrated measurements are available. This should be explained here.

For the Mie channel I found the presentation here to be misleading. The text and Fig. 3 first had lead me to believe that in the case of the Mie channel, that the data yieled by the device was a spectrum such as that illustrated in Fig.3. It took some time and iterating between the two documents, to realize that that for the Mie channel, that the curve shown in Fig. 3 corresponds only to the central position of the Fizeau wedge !

Only after reading through the L1 and L2 ATBDs I understood that there are 16 different spectral channels available. Further, the response of each channel is the result of integrating the spectrometer output image along the different columns (corresponding to wavelength shift). Since the central wavelength varies as a function of Fizeau wedge position, the measurement will consist of the INTEGRATED filter spectral response (e.g. as shown in the bottom right-panel of Fig. 3) with the center frequency shifted according to its position along the CCD rows.

I understand that the author's would likely desire to keep the explanation concise, however, the presentation here really needs to be more detailed and accurate ! It did cost me some time to understand what was being shown here and how the instrument really functioned and I am sure this would also hold true to many other readers in the general community.

We understand that this description is not enough. We now aim to better introduce the interferometer design and the L1B product that is used as input to the L2A. We also added a better

description of the calibration coefficients and how they model the transmission of each type of spectrum, Rayleigh or Mie, through each channel. Channels which are also (somewhat confusingly) named Rayleigh or Mie.

Page 6: Lines 104-109
* * *
The description of the "MCA" is likely incomprehensible to anyone not intimately involved with the data processing itself ! What does "some sort of cross-talk correction" mean ? What is the L1B-derived scattering ratio ?

This should be easier to understand with a clearer L1B introduction now. We removed "some sort of"

Either provide more details about the MCA (even references to the appropriate sections of the publicly available ATBDs would help) or, if it is deemed not essential, to the paper just skip it.

The same general comments apply to the description of the ICA.

We added that we mention the ICA and group product for completeness and do not intend to describe it more in detail. The longer paragraph about them was removed.

We specifically added that we do not recommend to use them. As we describe the data released with v3.12, we think this might be a useful indication to the users.

Page 6: Lines 110
* * *
"At last.." ==> "Lastly,.."

Corrected

Section 2.2.1
* * *
See my later comment (Page 13: Line 252)

It would be useful for the general reader if it were to be explained what advantages (or disadvantages) the SCA method have compared to the usual method of determining extinction by calculating the log-derivative of the Rayleigh ATB profile ? Off hand. I can think that the low vertical resolution bins dealt with here may be a factor. Is this correct ?

Please see the response to the other associated comment.

Are any multiple-scattering considerations taken into account in the retrieval. It looks like they are not. Do you expect this to have any impacts on e.g. cirrus cloud retrievals ?

Multiple scattering is not taken into account in our algorithms. It is usually expected that the extremely narrow field of view of ALADIN is enough to avoid strong multiple scattering influence on the observations.

**Page 6: Line 118-119**

\_\_\_\_\_

"concision" is rarely used in modern English. I suggest "brevity" or "conciseness".

Sorry, that was a Gallicism!

The sentence is awkward: I suggest something like:

For the sake of brevity, only an outline of the SCA algorithm is presented here. Only the main features of the algorithm, necessary to understand the subsequent sections, are covered."

We adopted the reviewer's suggestion.

Page 7: Line 129

\_\_\_\_\_

Delete the "(dR(z)=R'(z)dz)" It is trivial and does not add anything to the presentation.

It has been deleted.

Page 7: Line 144
* * *
"..equation (2)" ==> "..equations (1) and (2)."

Changed according to suggestion

Page 8: Line 180

\_\_\_\_\_

"..thermal constraint on the primary mirror..." does not make any sense here. Do the authors mean "thermally induced distortion" or "thermal strain" ?

Yes, we changed the phrasing to "thermally induced distortion".

What is meant by "orbit phase" ? Do the authors mean the "orbit position" ? Does the distortion vary predictably along the orbit or is function of the e.g. solar background ?

We changed "orbit phase" to "position along the orbit".

The distortion seems to vary with the up-going thermal infrared radiation from the Earth.

Line 183: "...called the Instrument....(IRC) mode,..."

We added "mode"

Line 184" "...target with negligible Doppler shift due to the nadir pointing."

We changed the sentence according to the reviewer's suggestion

Page 9: Line 190
* * *
(Also relevant to Eqns. 5 and 6) What is the maximum height given by the AUX\_MET product. Is there any account given to the Rayleigh transmission between the top to the AUX\_MET product and the top-of-atmosphere ?

The AUX\_MET reaches ~80 km high, we use it to simulate optical properties of the molecular atmosphere up to this altitude, but do not account for anything higher than that.

Page 9: Line 198
* * *
"Constraints" ... see my comment (Page 8: Line 180)

changed to distortions

Page 9: Line 210:

It would be useful if the authors could elaborate

It would be useful if the authors could elaborate on this point a bit. For example, what order of magnitude error do they believe background aerosol levels may have on the accuracy of the calibration?

This is a difficult question. The L1B scattering ratio is used qualitatively, and its precision would not be sufficient to try to correct for the impact of particles on the calibration. If we trust the SR values, the absolute maximum increase in signal would be 16 %. In reality, this is probably only a few percent.

We now mention that "... might overestimate the radiometric coefficients by a few percent. Future work will investigate this potential source of bias."

Page 10: Line 215
* * *
See my comment above (Page 6: Lines 104-109). To the general reader the "L1B scattering ratio" is a meaningless term unless you explain it !

We added a section about the input data of the algorithms as section 2.2 of the revised paper.

Page 11: Lines 223-229
* * *
How is the lidar ratio chosen? Is it fixed or does it vary with altitude, latitude etc..

The MCA was designed to use a climatology of aerosols lidar ratio. However, the MCA doesn't have a capability to distinguish between clouds and aerosols. in practice, such a climatology isn't used and a fixed lidar ratio is used. It is currently of 1/0,07, i.e. around 14.

Section 3.1

It would be useful if the magnitude of the results of only measuring the co-polar return was discussed !

We added a discussion about the measured lidar ratios.

Page 12: Line 245:

"Designed as a wind lidar, ALADIN does not have the ability to measure depolarization". This sentence(along with the text that follows it) implies that this wind lidar do not (can not?) measure depolarization. Is this true in general or only for the specific design of ALADIN ? What design constraint has lead to ALADIN not detecting the co-polar return.

We mention that ALADIN is a wind lidar as a way to underline that aerosol studies where not in the initial scope of the mission. The main technical constraint is then that studying depolarisation was not part of the objectives of the mission.

This is the new sentence:

"Designed as a wind lidar, ALADIN was not initially aimed at observing aerosol optical properties in detail. Under these requirements, it was not fitted with the ability to measure depolarization."

Also, ALADIN transmits and recieves circularly polarized radiation NOT linearly polarized !

The polarization of the beam at various stages inside the instrument has been better described. We hope that the text is now clearer and that a curious reader can find enough information with the schematic of the instrument in the Science Report. Below is the modified paragraph:

"The UV laser beam is linearly polarized at the laser output. It goes through a quarter-wave plate (Fig. 4.13 in ESA 2008) before being routed towards the telescope and is thus transmitted towards the atmosphere with a circular polarization. On the way back, backscattered light goes again through the quarter-wave plate. The circularly polarized light that was transmitted might come back elliptically polarized in the case it was backscattered by depolarizing targets. After going through the quarter-wave plate is a mix of linearly polarized light, along the same direction as the transmitted light (co-polar) or along the perpendicular direction (cross-polar). The beam then reaches a polarized beam splitter. The co-polar light is routed towards the interferometers, while the cross-polar light is routed back towards the laser and is lost for the analysis. This means that, in order to compare Aeolus observations of backscatter coefficient and lidar ratio to other instruments, only the co-polar component must be considered."

**Page 13, Section 3.2**
* * *
The concept of the relationship between the extinction profile and the log-derivative of the Rayleigh attenuated backscatter profile is used throughout this section. From a mathematical view-point, it is certainly true that any approach to retrieving the extinction solely using the molecular backscatter profile (either explicitly or implicitly) involves computing the log derivative of the attenuated backscatter profile. This must be true also of the SCA approach briefly described in Section 2.2.1. It would be useful to guide the reader with regards to this point. For example, outlining how the SCA approach is related to the standard log-derivative method for retrieving extinction would be useful !

In the L2A ATBD, the approximation applied to molecular extinction In Eq. 6.34 is not applied to to particle extinction in Eq. 6.35. This is because ALADIN range bins are thicker than for most lidar systems and the total extinction from particle within a given range bin becomes large. The use of function H(x) rather than simply e-x is more "correct". We wrote the following paragraph:

"Extinction can be calculated in a simple way from the molecular backscatter, or more precisely, from its derivative. The SCA is very similar to the classical log-derivative algorithms but the thickness of ALADIN range bins (up to 2 km) mean that the particulate optical thickness (\$\alpha\_p \* \Delta R\$) can be large and the approximation used for the molecular extinction (Eq. 6.34 in \ cite{flamant\_aeolus\_2021}) cannot be used for \$\alpha\_p\$. This is why we later need to inverse function H rather than simply derive the logarithm of the attenuation of the Rayleigh signal. As a side note, this refinement is also the reason why the adjective "correct" was added to the name of the algorithm."

Page 16: Line 308
* * *
"..are out of the graphics.." ==> "..are off scale in Figure 9..."

This was corrected according to suggestion.

Page 16: Lines 229-330:
* * *
The naming of the instruments and the platforms they are on are all conflated here ! I suggest, for example, CALIOP on board the NASA/CNES CALIPSO platform.

We changed the naming to the following format: "INSTRUMENT aboard Funding institution PLATFORM".;

Page 16: Lines 333:

\_\_\_\_\_

"...quality with.." ==> "..quality using.."

This was corrected according to suggestion.

Page 18: Fig. 11 Caption.
* * *
"..from 384 and 354nm spectral bands" ==> "derived using the 384 and 354nm spectral bands."

This was corrected according to suggestion.

Page 18: Line 345

\_\_\_\_\_

"This allows for the rejection of the low...."

Changed to "This allows for the rejection of the bins with low signal."

Page 19. Fig 1 and Section 4.2 in general.

\_\_\_\_\_

The figure is fine. However, it would be useful to also present the retrieved extinction as well as the Aeolus observed attenuated Mie and Rayleigh backscatter images. The absence of such images is conspicuous.

We added figure with the L1B useful signals and another figure with the attenuated backscatters.

Page 19, Lines 350-360.

There is a well-established relationship between the linear depolarization ratio and circular depolarization ratio that should hold for must circumstances. Given this it would be useful for the authors to give a quantitative number for the expected impact of the depolarization on the Aeolus measured lidar-ratio.

We improved the description of results, discussed the values and cited Wandinger et al. 2015.

Section 5:
* * *
Join the first two paragraphs.

Done

Can you please provide more detail connected with the points being made here ? For example:

-What type of new algorithms are being developed ?

We added a reference to:

Ehlers, F., Flament, T., Dabas, A., Trapon, D., Lacour, A., Baars, H., and Straume-Lindner, A. G.: Optimization of Aeolus Optical Prop-480erties Products by Maximum-Likelihood Estimation, preprint, Aerosols/Remote Sensing/Data Processing and Information Retrieval,https://doi.org/10.5194/amt-2021-212, https://amt.copernicus.org/preprints/amt-2021-212/, 2021b

and mentioned the work of the EarthCARE team to adapt their algorithms to Aeolus data.

**-Can you at least give a reference to the assimilation work ?**

**We added one**

Letertre-Danczak, J., Benedetti, A., Vasiljevic, D., Dabas, A., Flament, T., Trapon, D., and Mona, L.: Aerosol Assimilation of lidar data from505Satellite (AEOLUS) and Ground-based (EARLINET) instruments in COMPO-IFS., other, pico, https://doi.org/10.5194/egusphere-egu21-4799, https://meetingorganizer.copernicus.org/EGU21/EGU21-4799.html, 2021